# Response to short-term deprivation of the human adult visual cortex measured with 7T BOLD

**Paola Binda[1†], Jan W Kurzawski[2,3†], Claudia Lunghi[1,4], Laura Biagi[3], Michela Tosetti[3,5], Maria Concetta Morrone[1,3]***

[1]University of Pisa, Pisa, Italy; [2]Department of Neuroscience, University of Florence, Florence, Italy; [3]IRCCS Stella Maris, Pisa, Italy; [4]Département d'études cognitives, École normale supérieure, Laboratoire des systèmes perceptifs, PSL Research University, CNRS, Paris, France; [5]IMAGO Center, Pisa, Italy

**Abstract** Sensory deprivation during the post-natal 'critical period' leads to structural reorganization of the developing visual cortex. In adulthood, the visual cortex retains some flexibility and adapts to sensory deprivation. Here we show that short-term (2 hr) monocular deprivation in adult humans boosts the BOLD response to the deprived eye, changing ocular dominance of V1 vertices, consistent with homeostatic plasticity. The boost is strongest in V1, present in V2, V3 and V4 but absent in V3a and hMT+. Assessment of spatial frequency tuning in V1 by a population Receptive-Field technique shows that deprivation primarily boosts high spatial frequencies, consistent with a primary involvement of the parvocellular pathway. Crucially, the V1 deprivation effect correlates across participants with the perceptual increase of the deprived eye dominance assessed with binocular rivalry, suggesting a common origin. Our results demonstrate that visual cortex, particularly the ventral pathway, retains a high potential for homeostatic plasticity in the human adult.

DOI: https://doi.org/10.7554/eLife.40014.001

**\*For correspondence:**
concetta@in.cnr.it

[†]These authors contributed equally to this work

**Competing interests:** The authors declare that no competing interests exist.

## Introduction

To interact efficiently with the world, our brain needs to fine-tune its structure and function, adapting to a continuously changing external environment. This key property of the brain, called *neuroplasticity*, is most pronounced early in life, within the so called *critical period*, when abnormal experience can produce structural changes at the level of the primary sensory cortex (*Berardi et al., 2000*; *Hubel and Wiesel, 1970*; *Hubel et al., 1977*; *Wiesel and Hubel, 1963*). During development, occluding one eye for a few days induces a dramatic and permanent reorganization of ocular dominance columns (the V1 territory representing each eye) in favor of the open eye (*Berardi et al., 2000*; *Gordon and Stryker, 1996*; *Hubel and Wiesel, 1970*; *Hubel et al., 1977*; *Wiesel and Hubel, 1963*), while the deprived eye becomes functionally blind or very weak. These forms of structural plasticity have been documented in animal models, including non-human primates (*Gordon and Stryker, 1996*; *Kiorpes et al., 1998*; *Levi and Carkeet, 1993*; *Wiesel and Hubel, 1963*). A corresponding perceptual phenomenon known as amblyopia is observed in humans, and may result from exposing infants to monocular deprivation during the critical period, for example due to cataracts (*Braddick and Atkinson, 2011*; *Maurer et al., 2007*). In infants, even a partial deprivation produced by optical defects like astigmatism and myopia leads to a permanent acuity loss that cannot be compensated in adulthood, even after correction the optical aberrations (*Freeman and Thibos, 1975*) through Adaptive Optics (*Rossi et al., 2007*). Hebbian plasticity, endorsed by Long-Term synaptic

**eLife digest** The world around us changes all the time, and the brain must adapt to these changes. This process, known as neuroplasticity, peaks during development. Abnormal sensory input early in life can therefore cause lasting changes to the structure of the brain. One example of this is amblyopia or 'lazy eye'. Infants who receive insufficient input to one eye – for example, because of cataracts – can lose their sight in that eye, even if the cataracts are later removed. This is because the brain reorganizes itself to ignore messages from the affected eye.

Does the adult visual system also show neuroplasticity? To explore this question, Binda, Kurzawski et al. asked healthy adult volunteers to lie inside a high-resolution brain scanner with a patch covering one eye. At the start of the experiment, roughly half of the brain's primary visual cortex responded to sensory input from each eye. But when the volunteers removed the patch two hours later, this was no longer the case.

Some areas of the visual cortex that had previously responded to stimuli presented to the non-patched eye now responded to stimuli presented to the patched eye instead. The patched eye had also become more sensitive to visual stimuli. Indeed, these changes in visual sensitivity correlated with changes in brain activity in a pathway called the ventral visual stream. This pathway processes the fine details of images. Groups of neurons within this pathway that responded to stimuli presented to the patched eye were more sensitive to fine details after patching than before.

Visual regions of the adult brain thus retain a high degree of neuroplasticity. They adapt rapidly to changes in the environment, in this case by increasing their activity to compensate for a lack of input. Notably, these changes are in the opposite direction to those that occur as a result of visual deprivation during development. This has important implications because lazy eye syndrome is currently considered untreatable in adulthood.

DOI: https://doi.org/10.7554/eLife.40014.002

Potentiation and Depression (LTP/LTD) of early stage of cortical processing, underlies these changes in animal models and probably also in humans.

After the closure of the critical period, structural changes of V1 resulting from Hebbian plasticity are not typically observed (*Mitchell and Sengpiel, 2009*; *Sato and Stryker, 2008*). However, there is evidence that Hebbian plasticity can be restored in adult animal models under special conditions, associated with manipulation of the excitability of the visual cortex (*Fong et al., 2016*; *Frégnac et al., 1988*; *He et al., 2006*; *Maya Vetencourt et al., 2008*).

Besides Hebbian plasticity, other mechanisms can reshape primary visual cortex processing both within and outside the critical period. At the cellular level, there is evidence for homeostatic plasticity, which increases the gain of cortical responses following sensory deprivation; for example, after a brief monocular deprivation, the response gain of the deprived eye increases (*Maffei et al., 2004*). This is interpreted as an homeostatic response to preserve cortical excitability in spite of the synaptic depression produced by Hebbian plasticity, suggesting a close link between these two types of plasticity (*Maffei and Turrigiano, 2008*; *Turrigiano, 2012*) (*Mrsic-Flogel et al., 2007*; *Turrigiano and Nelson, 2004*).

In adult animal models and humans, there is clear evidence for both functional plasticity and for stability of the early sensory cortex (*Baseler et al., 2002*; *Baseler et al., 2011*; *Wandell and Smirnakis, 2009*). Functional changes have been observed with perceptual learning (*Dosher and Lu, 2017*; *Fahle and Poggio, 2002*; *Fiorentini and Berardi, 1980*; *Karni and Sagi, 1991*; *Karni and Sagi, 1993*; *Watanabe and Sasaki, 2015*), adaptation that, in some cases, may be very long-lasting, (*McCollough, 1965*), and short-term visual deprivation (*Binda and Lunghi, 2017*; *Kwon et al., 2009*; *Lunghi et al., 2015a*; *Lunghi et al., 2011*; *Lunghi et al., 2013*; *Mon-Williams et al., 1998*; *Zhang et al., 2009*; *Zhou et al., 2013*; *Zhou et al., 2014*). The effect of short-term deprivation in adults is paradoxical, boosting the perception of the deprived stimulus – opposite to the long-term deprivation effects during development. One of the first examples of short-term deprivation in adults is by *Mon-Williams et al., 1998*, who found that thirty minutes of simulated myopia (optical blur achieved by wearing a +1D lens) was followed by a transient improvement of visual acuity – opposite to the long-lasting acuity deficit produced by early onset myopia (*Rossi et al., 2007*).

Contrast attenuation for 4 hr leads to improved contrast discrimination thresholds and enhanced BOLD response in V1/V2 (*Kwon et al., 2009*). A few hours deprivation of one cardinal orientation leads to enhanced sensitivity to the deprived orientation (*Zhang et al., 2009*) – opposite to the reduced sensitivity to orientations deprived during development, for example due to astigmatism. Similarly, two hours of monocular contrast deprivation is followed by a transient boost of the deprived eye (*Binda and Lunghi, 2017*; *Lunghi et al., 2015a*; *Lunghi et al., 2011*; *Lunghi et al., 2013*; *Lunghi et al., 2015b*; *Zhou et al., 2013*; *Zhou et al., 2014*) and an enlargement of the deprived-eye representation at the level of V1 in non-human primates (*Begum and Tso, 2016*; *Tso et al., 2017*) – opposite to the amblyopia induced by monocular deprivation during the critical period. The mechanism supporting the perceptual boost of the deprived information could be either a form of homeostatic plasticity (like that observed in animal models), and/or a release of contrast adaptation for the deprived stimulus (*Blakemore and Campbell, 1969*; *Boynton et al., 1999*; *Gardner et al., 2005*; *Maffei et al., 1973*; *Movshon and Lennie, 1979*). Irrespective of the interpretation, the data clearly indicate that effects can be long-lasting or even permanent. For example, in patients with keratoconus (adult-onset corneal dystrophia, often monocular), best corrected visual acuity is worse than in emmetropic eyes, but it is better than predicted by the corneal dystrophy (*Sabesan and Yoon, 2009*; *Sabesan and Yoon, 2010*): when corneal aberrations of the keratoconic (KC) eyes are simulated in the emmetropic eyes, visual acuity is worse than in the KC eyes, demonstrating a permanent perceptual boost of the deprived information. Moreover, in adult amblyopes (*Lunghi et al., 2018*), short-term monocular deprivation (of the amblyopic eye) may lead to permanent partial recovery of acuity (of the amblyopic eye). This observation resonates with the idea – introduced in the context of work at the cellular level – that homeostatic plasticity and Hebbian plasticity may be fundamentally linked (*Maffei and Turrigiano, 2008*) and may open important new pathways for the therapy of amblyopia and, in general, for the rehabilitation of early-onset visual dysfunctions (*Legge and Chung, 2016*).

This possibility highlights the importance of understanding the neural substrates of short-term deprivation in adult humans. So far, monocular deprivation effects have been indirectly studied with MR spectroscopy (showing a GABA concentration change in the occipital cortex, *Lunghi et al., 2015b*) and Visual Evoked Potentials (showing a modulation of the early visual response components, *Lunghi et al., 2015a*). Indirect evidence also indicates that deprivation effects are not generalized but preferentially involve the parvocellular pathway – given that effects are more prominent and longer-lasting for chromatic equiluminant stimuli in humans (*Lunghi et al., 2013*), and strongest in macaques when deprivation mainly affects the parvocellular activity (*Begum and Tso, 2016*). Here we directly measure the changes in early visual cortical areas using 7T fMRI in adult humans, before and after two hours of monocular deprivation. Assessing the BOLD change and its selectivity to spatial frequency with a newly developed approach (conceptually similar to the population Receptive Field method, *Dumoulin and Wandell, 2008*), we demonstrate a change of ocular drive of BOLD signals in primary visual cortex, selective for the higher spatial frequencies and strongest along the ventral pathway, consistent with a stronger plasticity potential of the parvocellular pathway in adulthood.

## Results

### Monocular deprivation boosts V1 responses to the deprived eye and shifts BOLD ocular dominance

To investigate the visual modulation of BOLD signal by short term deprivation, we performed ultra-high field (UHF, 7T) fMRI during the presentation of high contrast dynamic visual stimuli, delivered separately to the two eyes, before and after 2 hr of monocular contrast deprivation (see schematic diagram in *Figure 1A*).

The reliability and high signal-to-noise ratio of our system allow us to obtain significant activations with only two blocks of stimulation (*Figure 1C* shows the profile of V1 BOLD response), thereby targeting the first 10 min after deprivation, when the perceptual effects are strongest (*Lunghi et al., 2011*; *Lunghi et al., 2013*). As shown in *Figure 1B*, the stimulation was sufficient to reliably activate most early visual areas (dashed lines outline ROIs limited by stimulus eccentricity, as detailed in the Materials and method).

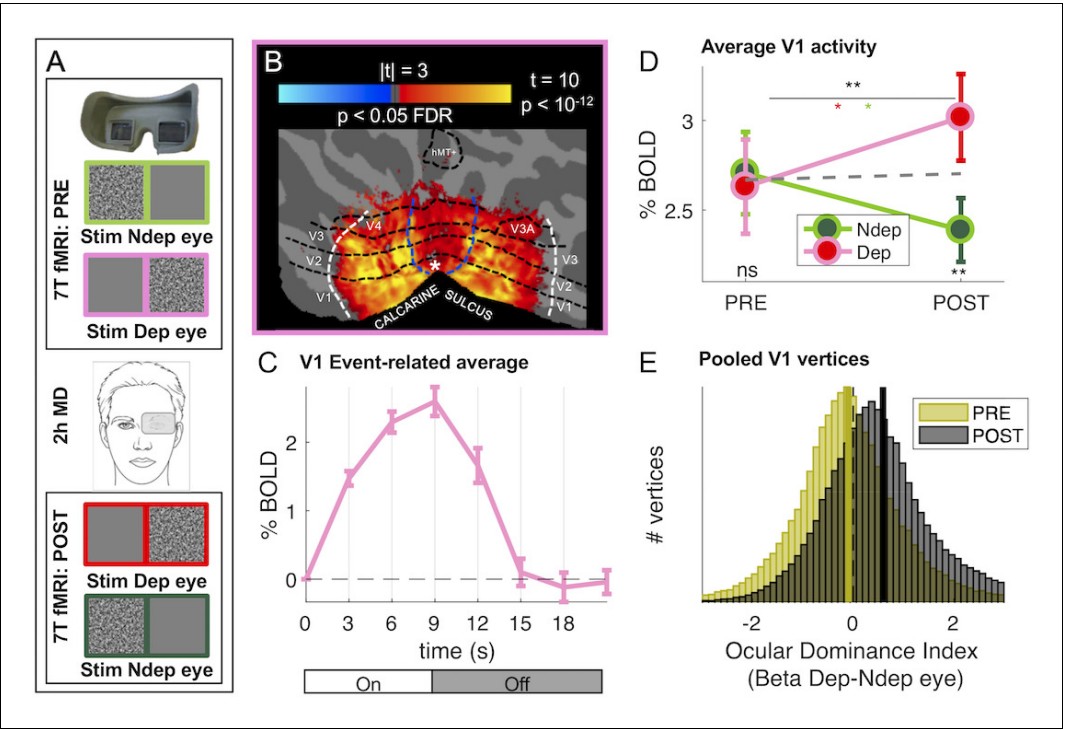

**Figure 1.** Monocular deprivation modulates 7T BOLD responses in early visual cortex. (**A**) Schematic illustration of the methods. The icons show a band-pass noise stimulus shown to either eye through the MR compatible goggles. Before and after the Pre- and Post-deprivation scans, outside the bore, we also measured binocular rivalry. (**B**) BOLD responses evoked by our band-pass noise stimulus with peak frequency 2.7 cycles per degree (cpd), presented in the deprived eye PRE-deprivation, mapped on the flattened cortical surface, cut at the calcarine sulcus. T-values are obtained by aligning GLM betas for each subject and hemisphere to a left/right symmetric template hemisphere, excluding vertices for which preferred eccentricity was not adequately estimated or smaller than 1 (the same criterion used for al analyses), then evaluating the distribution of betas in each vertex against 0 (one-sample t-test) and FDR correcting across the entire cortical surface. Black dashed lines show the approximate average location of the regions of interest V1 through MT, which were mapped on the individual subject spaces (see Materials and methods); white and blue lines represent the outer limits of the representation of our screen space (24 × 32 deg) and the foveal representation (≤1 deg, where eccentricity could not be mapped accurately) respectively. (**C**) BOLD modulation during the 3 TRs of stimulus presentation (from 0 to 9 s) and the following four blank TRs, for the 2.7 cpd noise stimuli delivered to the deprived eye before deprivation. The y-axis show the median percent BOLD signal change in V1 vertices relative to the signal at stimulus onset, averaged across subjects. Error bars give s.e. across participants. Note the small between-subject variability of the response (given that the response of each subject was computed for just two blocks of stimulation-blank). D: Average BOLD response to the band-pass noise stimulus with peak frequency 2.7 cpd, in each of the four conditions, computed by taking the median BOLD response across all V1 vertices then averaging these values across participants (after checking that distributions do not deviate from normality, Jarque-Bera hypothesis test of composite normality, all p > 0.06). The top black star indicates the significance of the ANOVA interaction between factors time (PRE, POST deprivation) and eye (deprived, non-deprived); the other stars report the results of post-hoc t-tests: red and green stars give the significance of the difference POST minus PRE, for the deprived and non-deprived eye respectively; bottom black stars give the significance of the difference deprived minus non-deprived eye before and after deprivation. *p < 0.05; **p < 0.01; ***p < 0.001; ns non-significant. E: Histograms of Ocular Drive Index: the difference between the response (GLM beta) to the deprived and non-deprived eye, computed for each vertex, separately before and after deprivation. Yellow and black lines give the median of the distributions, which are non-normal (logistic) due to excess kurtosis.

DOI: https://doi.org/10.7554/eLife.40014.003

The following figure supplements are available for figure 1:

**Figure supplement 1.** Effects of deprivation across the visual cortex Monocular deprivation had strong and opposite effects on the response to the 2.7 cpd stimulus in the two eyes.

DOI: https://doi.org/10.7554/eLife.40014.004

**Figure supplement 2.** Change of ocular preference after deprivation.

*Figure 1 continued on next page*

*Figure 1 continued*

DOI: https://doi.org/10.7554/eLife.40014.005

**Figure supplement 3.** Split-half reliability of the deprivation effect in V1.

DOI: https://doi.org/10.7554/eLife.40014.006

We measured the plasticity effect by comparing activity before/after deprivation in response to stimulation in the two eyes with low- and high-spatial frequency bandpass stimuli that differentially stimulate the magno- and parvocellular pathways (see *Figure 1—figure supplement 1* panels C-D for maps of responses to stimuli in both eyes, before and after deprivation). Consistent with prior evidence suggesting higher susceptibility to plasticity of the parvocellular pathway (*Lunghi et al., 2015a*; *Lunghi et al., 2011*; *Lunghi et al., 2015b*; *Lunghi and Sale, 2015*), we observe a strong effect of Monocular Deprivation on BOLD responses to stimuli of high spatial frequency (peak 2.7 cycles per degree, high-frequency cut-off at half-height 7.5 cpd). *Figure 1D* shows that the V1 response to the high spatial frequency stimuli presented in the left and right eye is nearly equal before deprivation ('PRE') (see *Figure 1—figure supplement 1*, panels C-D and *Figure 1—figure supplement 2*, panel A, mapping the difference between responses to the two eyes). However, after deprivation ('POST'), the response in the two eyes changes in opposite directions, with a boost of the BOLD response (measured as GLM Beta values, expressed in units of % signal change) of the deprived eye and a suppression of the non-deprived eye (see also *Figure 1—figure supplement 2*, panel B). This was formally tested with a two-way repeated measure ANOVA, entered with the mean BOLD responses across all vertices in the left and right V1 region, for the four conditions and each participant (*Figure 1D* show averages of this values across participants). The result reveals a significant interaction between the factors *time* (PRE, POST deprivation) and *eye* (deprived, non-deprived; interaction term $F(1,18) = 13.80703$, $p = 0.00158$; the result survives a split-half reliability test: see *Figure 1—figure supplement 3*).

Fig. 1E confirms these findings with an analysis of the aggregate subject data, obtained by pooling all V1 vertices across all subjects. For each vertex, we defined an index of Ocular Dominance computed as the difference of BOLD response to the deprived and non-deprived eye. This index is not to be confused with the anatomical arrangement of vertices with different eye preference that define the ocular dominance columns (*Cheng et al., 2001*; *Yacoub et al., 2007*), that cannot be directly imaged with voxel size of 1.5 mm. However, at this low resolution, each voxel is expected to average signals from a biased sample of ocular dominance columns leading to an eye preference of that particular voxel (the Ocular Dominance index in *Figure 1E*).

Before deprivation, the Ocular Dominance index is symmetrically distributed around zero, indicating a balanced representation of the two eyes before deprivation (yellow distribution in *Figure 1E*). After deprivation (black distribution in *Figure 1E*), the Ocular Dominance distribution shifts to the right of 0, indicating a preference for the deprived eye (non-parametric Wilcoxon sign-rank test comparing the PRE and POST Ocular Dominance medians, $z = 115.39$, $p < 0.001$).

In principle, the boost of responses to the deprived eye seen in *Figure 1D* could be produced by enhancing the response of vertices that originally preferred the deprived eye (without shifting ocular dominance) or by changing Ocular Dominance of vertices that originally preferred the non-deprived eye, driving them to prefer the deprived eye. The shift of the Ocular Dominance histogram in *Figure 1E* is more compatible with the latter case, implying a recruitment of cortical resources for the representation of the deprived eye. To investigate this further, we monitored the final POST-deprivation Ocular Dominance of individual vertices that, PRE-deprivation, preferred the deprived eye (yellow half distribution in *Figure 2B*). The majority of vertices continue to prefer the same eye before and after deprivation. The median Ocular Dominance is significantly larger than 0 both PRE and POST (Wilcoxon sign-rank test, $z > 101.54$, $p < 0.0001$ in both cases) and the correlation between Ocular Dominance indices before and after deprivation is strong and positive (Pearson's R (32236) = 0.22 [0.21–0.23], $p < 0.0001$). Note that a completely random reassignment of Ocular Dominance after deprivation would have produced a histogram centered at 0 and no correlation between Ocular Dominance indices PRE- and POST deprivation. This is not consistent with the results of *Figure 2B*, which thereby provide evidence that our estimates of Ocular Dominance before and after deprivation are congruent, even though they were collected in different fMRI sessions separated by 2 hr. In addition, the distribution of Ocular Dominance after deprivation is well predicted

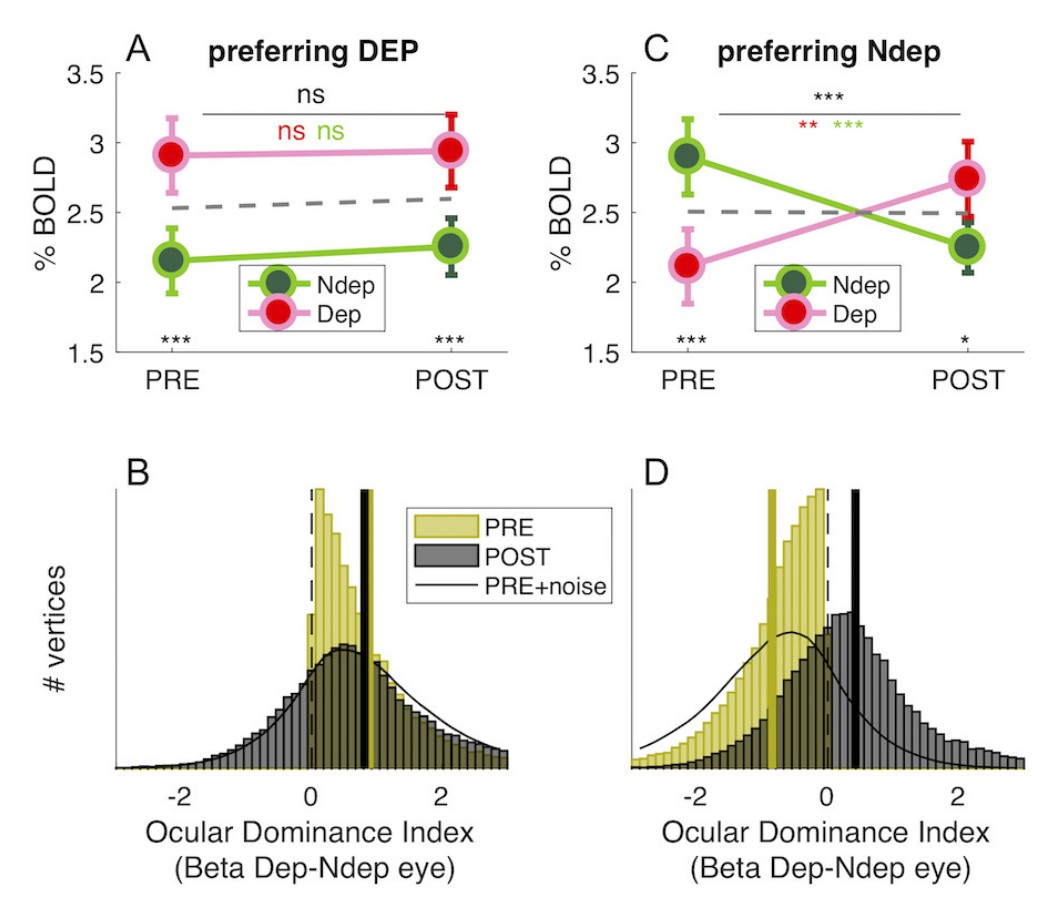

**Figure 2.** Monocular deprivation shifts 7T BOLD Ocular Dominance in V1. (A) and (C) Average BOLD responses with the same conventions as in *Figure 1D* but analysing data from two sub-regions of V1. (A) only vertices that, before deprivation, respond preferentially to the deprived eye. (C) only vertices that, before deprivation, respond preferentially to the non-deprived eye. (B) and (D) Histograms of Ocular Dominance Index (as for *Figure 1E*), in the two sub-regions of V1, computed before and after deprivation. The black curve simulates the result of adding random noise to the distribution obtained before deprivation; only in B does this approximate the distribution observed after deprivation.

DOI: https://doi.org/10.7554/eLife.40014.007

by adding only a small amount of noise to the original half distribution (Gaussian noise with 0.12 standard deviation, black line), suggesting that these vertices were largely unaffected by monocular deprivation. This is also supported by the repeated measure ANOVA of individual subject data (*Figure 2A*), revealing a strong main effect of eye (F(1,18) = 48.28901, $p < 10^{-5}$): the response to the deprived eye is stronger than the non-deprived eye, both before deprivation (due the selection, t(18) = −8.616, $p < 10^{-5}$), and after deprivation (t(18) = −4.281, $p < 10^{-5}$), with no effect of time and no *time × eye* interaction (all F(1,18) = 0.20429, p > 0.5).

A completely different pattern is observed for the vertices originally preferring the non-deprived (yellow half-distribution in *Figure 2D*). Here the distribution of Ocular Dominance clearly shifts after deprivation; the median moves from significantly negative before deprivation (Wilcoxon sign-rank test, z = −175.97, p < 0.0001) to significantly positive after deprivation (Wilcoxon sign-rank test, z = 64.46, p < 0.0001), implying a shift of dominance in favor of the deprived eye. Again, this is not consistent with a random reassignment of Ocular Dominance after deprivation, which predicts a distribution centered at 0. Contrary to *Figure 2B*, the POST- Ocular Dominance distribution cannot be predicted by injecting Gaussian noise to the PRE- Ocular Dominance distribution (black line, 0.12 standard deviation like for *Figure 2B*): for these vertices, there is a shift of Ocular Dominance with

short term monocular deprivation. This is confirmed with the repeated measure ANOVA (*Figure 2C*), where the *time × eye* interaction is significant (F(1,18) = 44.82812, p < 10⁻⁵), implying a different modulation PRE and POST deprivation. In addition and crucially, POST-deprivation BOLD responses to the deprived eye are significantly larger than POST-deprivation responses to the non-deprived eye (t(18) = −2.775 p = 0.012; whereas, by selection, the opposite is true before deprivation: t(18) = 12.034, p < 10⁻⁵).

In summary, Ocular Dominance before deprivation defines two similarly sized sub-regions of V1 vertices (44.58 ± 5.38% and 55.42 ± 5.38% of analyzed V1 vertices; 44.84 ± 5.12% and 55.16 ± 5.12% of all V1 vertices) with radically different behaviors that are not consistent with an artifact induced by vertex selection. The sub-region that originally represents the deprived eye does not change with deprivation; the sub-region that originally represents the non-deprived eye is rearranged with deprivation, as a large portion of vertices turn to prefer the deprived eye.

If plasticity were not eye-specific and/or we failed to match our V1 vertices before/after deprivation, we would expect that splitting the distribution of V1 ocular dominance generates opposite effects in the two subpopulations: vertices preferring the deprived eye before deprivation should swap to prefer the other eye, mirroring the effect seen in the vertices preferring non-deprived eye. This is not seen, implying that we did successfully match vertices across the 2 hr of deprivation and that the selective Ocular Dominance shift, observed for about half of our vertices, is not an artifact.

We also measured the perceptual effects of short-term monocular deprivation effects using Binocular Rivalry, just before the PRE- and POST-deprivation fMRI sessions. In line with previous studies (*Binda and Lunghi, 2017*; *Lunghi et al., 2015a*; *Lunghi et al., 2011*; *Lunghi et al., 2015b*; *Lunghi and Sale, 2015*), short-term monocular contrast deprivation induced a 30% increase of phase

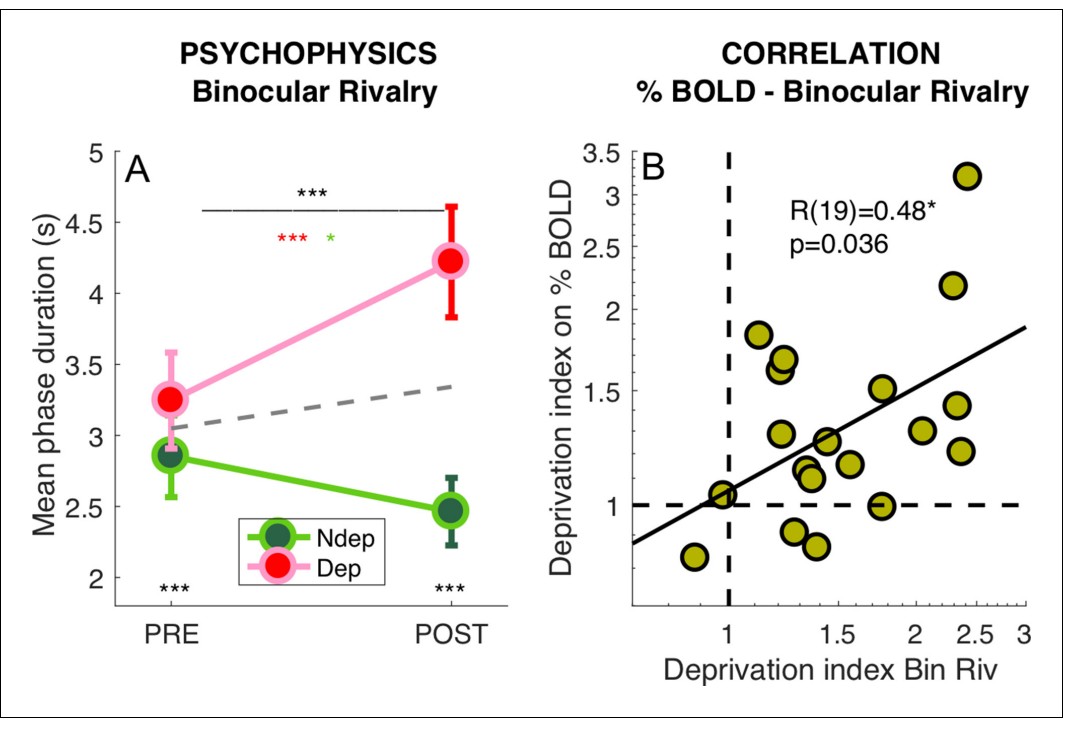

**Figure 3.** Deprivation effects on BOLD and on psychophysics are correlated. (A) Effect of deprivation on Binocular Rivalry dynamics. Average phase duration for the deprived and non-deprived eye, before and after deprivation, same conventions as in *Figure 1D*. Mean phase duration distributions do not deviate from normality (Jarque-Bera hypothesis test of composite normality, all p > 0.171) (B) Correlation between the deprivation index (the POST to PRE- ratio for the deprived eye divided by the same ratio for the non-deprived eye, *Equation 6* in Materials and method) computed for the binocular rivalry mean phase duration and for the BOLD response to our band-pass noise stimulus with peak frequency 2.7 cpd. Text insets show the Pearson's correlation coefficient and associated p-value..

DOI: https://doi.org/10.7554/eLife.40014.008

duration for the deprived eye (POST to PRE-deprivation ratio: 1.31 ± 0.30) and a 15% decrease of phase duration for the non-deprived eye (ratio: 0.86 ± 0.30), producing a significant *time × eye* interaction (*Figure 3A*, repeated measure ANOVA on the mean phase durations for each participant, interaction: $F(1,18) = 23.56957$, $p = 0.00013$). This effect size is similar to that measured in recent experiments using the same paradigm, but letting subjects continue normal activity during the 2 hr of monocular deprivation (*Lunghi et al., 2011*; *Lunghi et al., 2015b*; *Lunghi and Sale, 2015*). This indicates that the prolonged high contrast stimulation delivered for retinotopic mapping to the non-deprived eye during the first ~30 min of deprivation did not modulate the deprivation effects.

We defined a psychophysical index of the deprivation effect ($DI_{psycho}$) by using *Equation. 6* in Materials and methods section, where the POST to PRE-deprivation ratio of phase durations for the deprived eye, is divided by the same ratio for the non-deprived eye. Values larger than one imply a relative increase of the deprived eye phase duration, that is the expected effect; a value less than 1 indicates the opposite effect and a value of 1 indicates no change of mean phase duration across eyes. All but two subjects have values larger than 1, indicating a strong effect of deprivation. However, the scatter is large with values ranging from 0.7 to 3, suggesting that susceptibility to visual plasticity varies largely in our pool of participants. Capitalizing on this variability, we tested whether the size of the psychophysical effect correlates with the BOLD effect across participants. Using the same *Equation 6* to compute the deprivation effect on BOLD responses ($DI_{BOLD}$), we observed a strong correlation between the effect of monocular deprivation on psychophysics and BOLD (shown in *Figure 3B*). Subjects who showed a strong deprivation effect at psychophysics ($DI_{psycho} > 2$) also showed a strong deprivation effect in BOLD responses ($DI_{BOLD} = 1.85 ± 0.42$). Given that the psychophysics was measured only for central vision and at two cpd stationary grating, whereas BOLD responses were pooled across a large portion on V1 and were elicited using broadband dynamic stimuli, the correlation suggests that the psychophysical effect may be used as a reliable proxy of a general change of cortical excitability, which can be measured by fMRI.

## Monocular deprivation shifts BOLD spatial frequency tuning for the deprived eye

The BOLD measure we use here gives us the chance to measure the effect of Monocular Deprivation across spatial frequencies and as function of eccentricity. We used five band-pass noise (1.25 octaves half-width at half maximum) stimuli with peak spatial frequency selected to have a complete coverage of spatial frequencies from 0.03 to 12.5 cpd (see *Figure 4—figure supplement 1*). In contrast with the strong and reliable plasticity of responses to the high spatial frequency stimulus (peaking at 2.7 cpd, *Figures 1–3*), we find that the plasticity effect is absent at low spatial frequencies (interaction index for the highest spatial frequency stimulus: 0.70 ± 0.19; for the lowest spatial frequency stimulus: 0.16 ± 0.15; paired t-test $t(18) = -3.441$, $p = 0.003$).

Thus, monocular deprivation produces a change of the spatial frequency selectivity of the V1 BOLD response. Before deprivation, the BOLD response shows a broad band-pass selectivity for our stimuli, with a preference for the stimulus peaking at intermediate spatial frequencies, between 0.4 and 1.1 cpd, and a slight attenuation at higher spatial frequencies, similar for the two eyes (*Figure 4A*). After deprivation (*Figure 4B*), the non-deprived eye shows similar selectivity and an overall decrease of responses. For the deprived eye, the shape of the curve changes: from band-pass to high-pass, implying that the enhancement affects primarily the higher spatial frequencies.

To model this effect, we assume that each vertex on the cortical surface subtends a multitude of neuronal channels, each with narrow tuning for spatial frequency and collectively spanning a large range of spatial frequencies – an approach conceptually similar to the population Receptive Field model for retinotopic mapping (*Dumoulin and Wandell, 2008*). Independently of the exact bandwidth and peak preference of the neuronal population contributing to the final BOLD selectivity, we find that the shape of all these curves is captured with a simple one-parameter model: what we term the population tuning for Spatial Frequency. This is given by a Difference-of-Gaussians (DoG) function with one free parameter, the spatial constant (while the excitation/inhibition spatial constant ratio is fixed; see *Equation 4* in the Materials and method and curves in *Figure 5—figure supplement 1*). The free parameter sets the high spatial frequency cut-off at half-height of the filter. The continuous lines in *Figure 4* show how the model fits the grand-average of V1 responses, with best fit cut-off around five cpd similar for all conditions except for the POST-deprivation deprived eye,

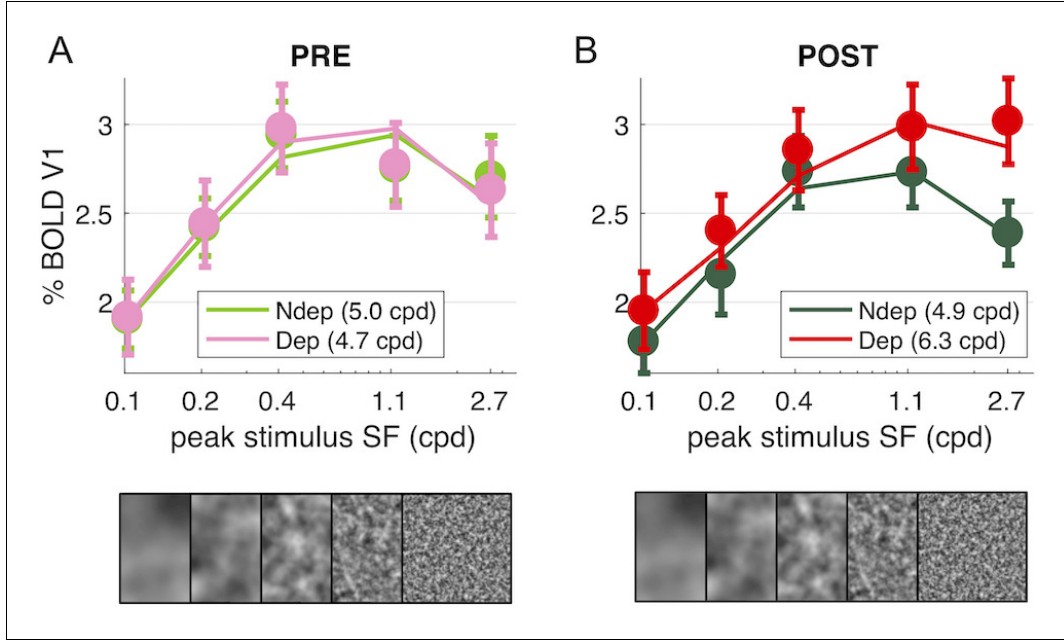

**Figure 4.** Deprivation affects spatial frequency selectivity in V1. V1 BOLD responses to all five of our band-pass noise stimuli (with peaks at 0.1, 0.2, 0.4, 1.1 and 2.7 cpd, see spectra in *Figure 4—figure supplement 1*); (**A**) response to stimuli in either eye, before deprivation; (**B**) response to stimuli in either eye, after deprivation. Responses are computed as medians across all V1 vertices (like in *Figure 1D*), averaged across subjects (error bars report s.e.m.). Continuous lines show the response of the best-fit population Spatial Frequency tuning (with the one parameter, the high spatial frequency cut-off, indicated in the legend), estimated by applying to the average V1 BOLD response the same model used to predict individual vertex responses (fitting procedure illustrated in *Figure 5—figure supplement 1*)..

DOI: https://doi.org/10.7554/eLife.40014.009

The following figure supplement is available for figure 4:

**Figure supplement 1.** Bandpass noise stimuli.

DOI: https://doi.org/10.7554/eLife.40014.010

---

where the cut-off is 6.2 cpd (single vertex examples are given in *Figure 5—figure supplement 1* panels C-I). The DoG equation has been successfully used in previous studies to model CSF and neural responses at variable stimulus parameters for example illumination levels (*Enroth-Cugell and Robson, 1966*; *Hawken et al., 1988*), validating this equation for modeling the overall selectivity of large neuronal ensembles.

Using this model to analyze single vertex responses, we evaluated the best-fit spatial frequency cut-off of the neural population contributing to the vertex BOLD response (see details in the Materials and method and *Figure 5—figure supplement 1* panels A-C; briefly, we used the DoG model to predict the response elicited by our five band-pass noise stimuli in populations with different spatial frequency selectivity, that is filters with different cut-off; we then found the cut-off value that maximizes the correlation between the predicted responses and the observed BOLD responses). We used this procedure to fit BOLD responses in each of our four conditions, estimating spatial frequency selectivity in individual vertices in each condition: separately for the two eyes, PRE/POST deprivation. Before deprivation, the spatial frequency cut-off decays with eccentricity as expected. *Figure 5A* maps both eccentricity (pRF eccentricity estimates from a separate retinotopic mapping scan) and spatial frequency cut-off values, obtained by fitting responses to the deprived eye, before deprivation (averaged across hemispheres and subjects). The cut-off is around 16 in the para-fovea (eccentricity around 1.5 deg) and down to four in the periphery (eccentricity around 8 deg). This relationship between eccentricity and spatial frequency preference is consistent with previous fMRI results (*D'Souza et al., 2016*; *Henriksson et al., 2008*) and with psychophysics (*Rovamo et al., 1978*). The model captures well the selectivity of an example V1 vertex (*Figure 5B*,

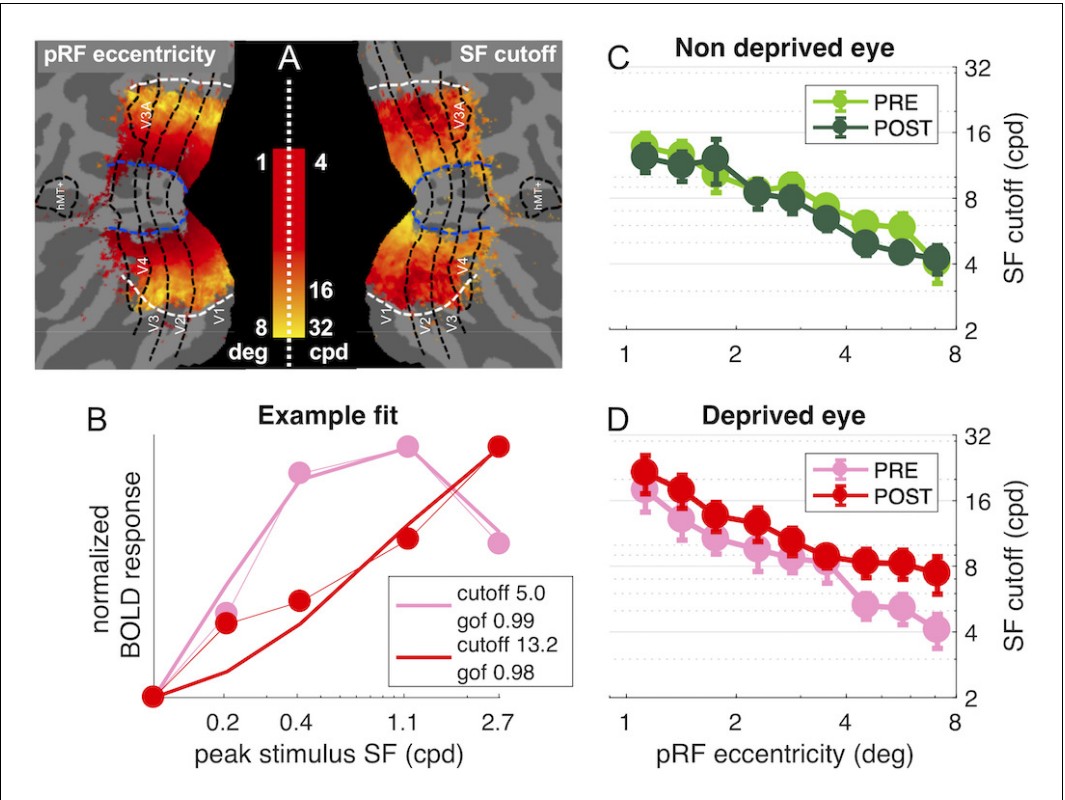

**Figure 5.** population Spatial Frequency Tuning in V1. (**A**) Maps of pRF eccentricity and best fit spatial frequency cut off (for the deprived eye before deprivation) after aligning the parameter estimates for all hemispheres to a common template and averaging them across subjects and hemispheres, after excluding vertices for which the average preferred eccentricity was not adequately estimated or smaller than 1 (the same exclusion criteria used for analyses). (**B**) Predicted and observed BOLD activity in one example vertex, elicited in response to our bandpass noise stimuli in the deprived eye PRE (pink) and POST deprivation (red), with best fit spatial frequency cut off (reported in the legend). (**C-D**) Best fit spatial frequency cut-off, averaged in sub-regions of V1 defined by pRF eccentricity bands, and estimated separately for the two eyes and PRE/POST deprivation.

DOI: https://doi.org/10.7554/eLife.40014.011

The following figure supplement is available for figure 5:

**Figure supplement 1.** population Spatial Frequency Tuning estimation.

DOI: https://doi.org/10.7554/eLife.40014.012

goodness of fit better than 0.9), sampled from the mid-periphery (3.4 deg) for the deprived eye, both before and after deprivation. The spatial frequency cut-off after deprivation shifts to higher values, increasing (in this example) by about a factor of three. *Figure 5C–D* shows that this behavior is systematically observed across V1 vertices, but only for the deprived eye. Here the average cut-off is plotted as function of eccentricity, and the roll-off is consistent with the map in *Figure 5A*. For the non-deprived eye, there is no effect of deprivation on spatial frequency selectivity (*Figure 5C*). In contrast, for the deprived eye (*Figure 5D*), there is a shift towards preferring higher spatial frequencies, at all eccentricities, which is captured by an increased value of the cut-off frequency parameter leading to an increased acuity of the BOLD response to the deprived eye.

Note that the change of spatial frequency selectivity for the deprived eye is most evident at eccentricities of 4 deg and higher (see *Figure 5D*), where vertices have peak sensitivity at mid-to-low spatial frequencies before deprivation. In the fovea, where many vertices already prefer the highest spatial frequency stimulus before deprivation, our fitting procedure is likely to underestimate the change of spatial frequency selectivity. Importantly, the spatial frequency selectivity for the non-deprived eye is unchanged at all eccentricities, corroborating the eye and stimulus specificity of the short-term monocular deprivation effect. These findings are consistent with maps in *Figure 1—*

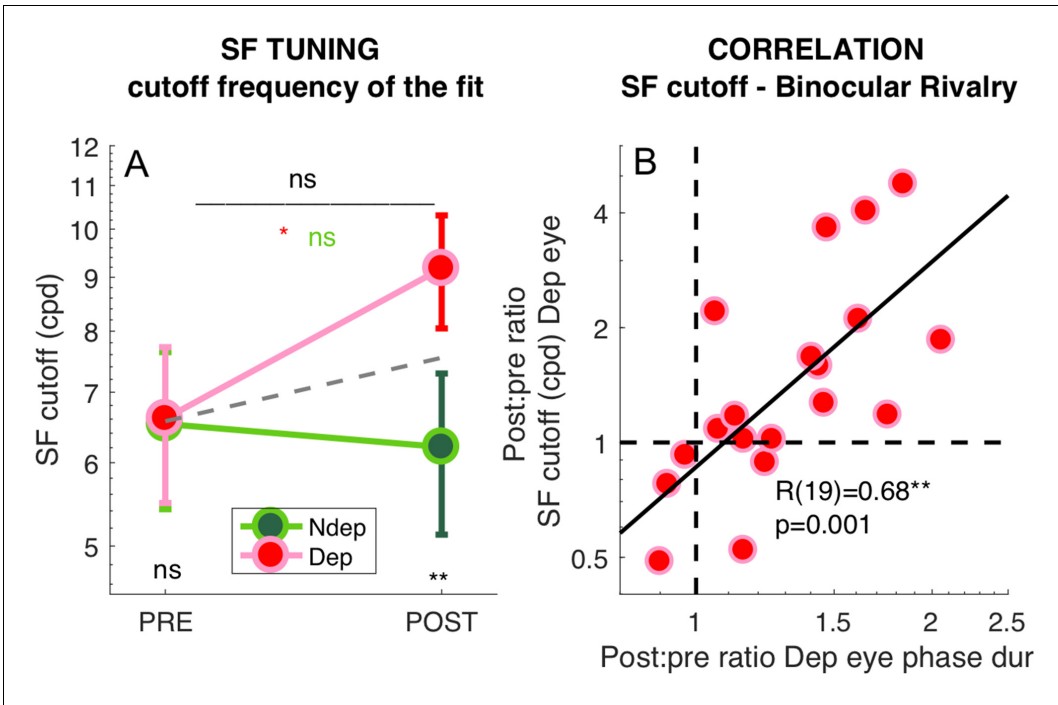

**Figure 6.** Deprivation effects on the deprived eye population Spatial Frequency Tuning and binocular rivalry phase duration are correlated. (**A**) Effect of deprivation on spatial frequency cut off values. Average cut-off across all V1 vertices (pooled across eccentricities) for the deprived and non-deprived eye, before and after deprivation, same conventions as in *Figure 1D*. Distributions of the log-values do not deviate from normality (Jarque-Bera hypothesis test of composite normality, all p > 0.285). (**B**) Correlation between the POST/PRE ratio (*Equation 7* in the Materials and methods) computed for the binocular rivalry mean phase duration and for the spatial frequency cut off for the deprived eye.

DOI: https://doi.org/10.7554/eLife.40014.013

*figure supplement 1* panels C-D showing that deprivation effects are largely homogenous across all V1 eccentricities, with no obvious clustering of effects in the fovea or in the periphery.

To test the significance of these effects, we pooled the best fit cut-off values from all selected V1 vertices across eccentricities and averaged them across participants (*Figure 6A*). The repeated measure ANOVA (performed on the log-transformed values, which are distributed normally as assessed by the Jarque-Bera test) shows no significant *time × eye* interaction (F(1,18) = 3.67607, p 0.07121) and non significant main effect of time (F(1,18) = 2.62546, p = 0.12255) but a significant main effect of eye (F(1,18) = 13.58079, p = 0.00169). This is clarified by post-hoc t-tests revealing that the increase of spatial frequency cut-off for the deprived eye is significant (t(18) = −2.263, p = 0.036) whereas there is no significant change for the non-deprived eye (t(18) = 0.440, p = 0.665). Given that the *time × eye* interaction in the full V1 region is not significant, and to minimize noise contamination, we evaluated the effect of deprivation on spatial frequency cut-off at the individual level by a 'Deprived Eye Change (DepC$_{cutoff}$)' index (*Equation 7* in the Materials and method), that is taking the POST vs. PRE-deprivation ratio of the spatial frequency cut-off for the deprived eye alone. As this ratio varies widely across participants, over more than three octaves, we asked whether this variability correlates with our psychophysical probe of plasticity: binocular rivalry. We used the same *Equation 7* to index the psychophysical change of the deprived eye (DepC$_{psycho}$), the POST to PRE-ratio of mean phase duration for the deprived eye, and found a strong positive correlation (*Figure 6B*). POST-deprivation, the deprived eye shows an increase of mean phase duration (in binocular rivalry) and an increase of the spatial frequency cut-off (best fit of the BOLD responses): participants showing a stronger increase of phase duration, also showed a larger shift of selectivity towards higher spatial frequency. The correlation is consistent with the result of *Figure 3* showing

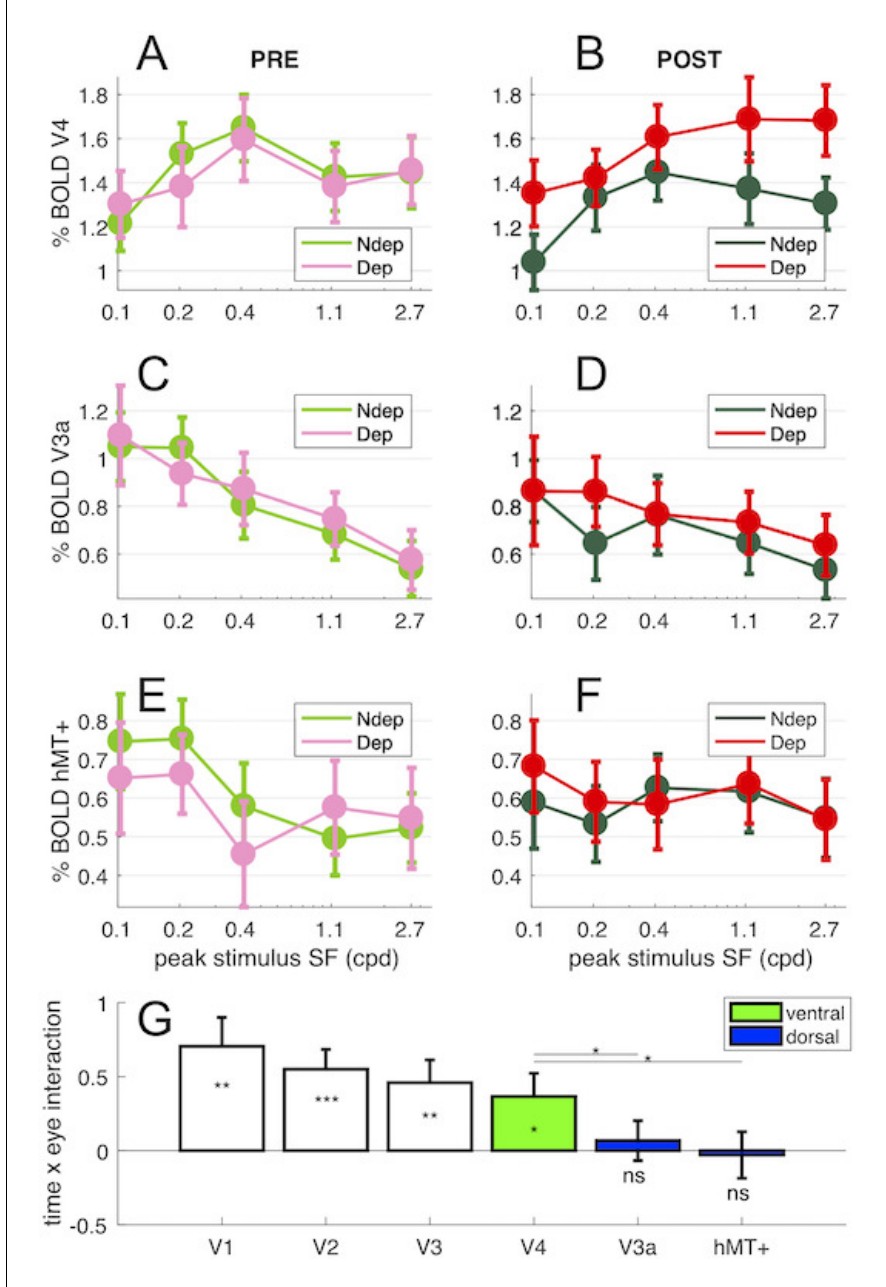

**Figure 7.** Deprivation effects are stronger in ventral than in dorsal stream areas. Panels (**A-B**) show V4 responses across spatial frequency stimuli presented to each eye (colored lines) before (A) and after deprivation; panels (**C-D**) show V3a responses and panels (**E-F**) show hMT+ responses. Each data point is computed by taking the median BOLD response across vertices in the region of interest for each stimulus and subject, then averaging across subjects (errorbar report s.e.m.). Panel (**G**) summarizes the effect of deprivation measured for the highest spatial frequency stimulus in the V1, V2, V3/VP, V4, V3a and hMT+ region of interest, computing the interaction term (POST-PRE difference of BOLD response for the deprived eye, minus the same value for the non-deprived eye) for individual participants and the 2.7 cpd stimulus. Values around 0 indicate no effect of deprivation and values larger than 0 indicate a boost of the deprived eye after deprivation. One-sample t-tests comparing this value against 0 give a p-value equivalent to that associated with the interaction term of the ANOVA (**Figure 1D**); the significance of the resulting t-value is given by the stars plotted below each errorbar. Stars plotted above the lines show the results of paired t-tests comparing interaction terms in V4 and V3a/hMT+. *** = p < 0.001; ** = p < 0.01; * = p < 0.05; ns = p ≥ 0.05. Green and Blue highlight the assignment of the higher tier areas to the ventral and dorsal stream respectively.

DOI: https://doi.org/10.7554/eLife.40014.014

that the enhancement of BOLD responses is correlated with the change of binocular rivalry and selective for the highest spatial frequency stimulus.

## Monocular deprivation affects BOLD responses in the ventral stream areas beyond V1

We measured the effect over the main extra-striate visual cortical areas. The selective boost of the deprived eye response to the high spatial frequency is as strong in V2 as in V1 (*Figure 1—figure supplement 1* and *Figure 7E*). The boost is present also in V3 and V4. In V4 the boost appears to be present also for lower spatial frequencies, but again only for the deprived eye (*Figure 7A–B*), possibly reflecting the larger spatial frequency bandwidth of V4 neurons compared to V1.

The results are very different for dorsal area V3a (*Figure 7C–D*) and hMT+ (*Figure 7E–F*), which do not show any significant change of responses in either eye at high spatial frequencies. Although the preferred response moves to lower spatial frequencies, consistent with a stronger input of the magnocellular pathway to the dorsal visual stream (*Henriksson et al., 2008*; *Singh et al., 2000*), the response to the highest spatial frequency stimulus is still strong and reliable in both V3a and hMT+. Note that the reliable BOLD estimates of *Figure 7* are computed after pooling vertices within the ROI and then averaging across subjects. However, the response of hMT+ evaluated at the individual vertex do not show significant activation (*Figure 1B*), probably reflecting more variable organization of activity within this ROI across subjects (*Smith et al., 2006*).

Fig 7G quantifies the effect of short-term monocular deprivation (using the ANOVA time x eye interaction term, which measures the eye-selective modulation of BOLD response after deprivation for the highest spatial frequency) across the main visual areas. The plasticity effect is strongest in V1, V2 and V3; it is still strong and significant in ventral area V4 ($t(18) = 2.41$ $p = 0.0270$), but it is absent in V3a and hMT+, where the time x eye interaction is not significantly different from 0 ($t(18) = 0.52$ $p = 0.6115$ and $t(18) = -0.19$ $p = 0.8513$ respectively). The plasticity effect in ventral area V4 is significantly stronger than in dorsal areas V3a and hMT+ ($t(18) = 2.39$, $p = 0.0278$ and $t(18) = 2.36$, $p = 0.0299$ for V4-V3a and V4-hMT+ respectively).

This result suggests a preferential involvement of the parvocellular vs. magnocellular pathway, leading to the differential plasticity effect in extra-striate visual areas of the ventral and dorsal pathway. Interestingly, the plasticity effect is robust in areas where the majority of cells are binocular (like V3 and V4), indicating that the effect does not require segregated representations of the two eyes (e.g. Ocular Dominance columns).

## Discussion

We demonstrate that two hours of abnormal visual experience has a profound impact on the neural sensitivity and selectivity of V1. BOLD activity across the V1 cortical region paradoxically increases for the eye that was deprived of contrast vision, and decreases for the eye exposed to normal visual experience.

The enhanced response to the deprived eye fits well with the concept of homeostatic plasticity, first observed in rodent visual cortex, both juvenile and adult (*Maffei et al., 2004*; *Mrsic-Flogel et al., 2007*; *Turrigiano and Nelson, 2004*), which is the tendency of neural circuits to keep the average firing rates constant in spite of anomalous stimulation (*Maffei and Turrigiano, 2008*; *Turrigiano, 2012*) (*Mrsic-Flogel et al., 2007*; *Turrigiano and Nelson, 2004*). More recently, similar observations have been made in the adult macaque V1 after two hours of monocular deprivation during anesthesia (*Begum and Tso, 2016*; *Tso et al., 2017*). The post-deprivation gain boost observed in the monkey is consistent with our observations of an increased BOLD response to the deprived eye. We also observe an antagonistic suppression of the non-deprived eye BOLD response; together, the two effects lead to a shift of ocular preference of individual vertices in favor of the deprived eye. However, this effect is only observed in those V1 vertices that responded preferentially to the non-deprived eye before deprivation. No change of ocular preference is seen in vertices that already prefer the deprived eye before deprivation, which maintain their eye-preference after deprivation. This pattern of results cannot be explained by an overall gain increase; rather, it is consistent with the idea that the representation of the deprived eye recruits cortical resources (which may or may not correspond to cortical territory), normally dedicated to the other eye.

A similar antagonist effect on the two eyes (boosting the deprived eye and suppressing the non-deprived eye) was also observed in the VEP responses after short-term monocular deprivation (*Lunghi et al., 2015a*), and could be implemented through a modulation of the excitatory/inhibitory circuitry. Regulation of the excitation/inhibition balance through GABAergic signaling is considered to be a key factor for cortical plasticity, including homeostatic plasticity (*Maffei and Turrigiano, 2008*). Interestingly, the involvement of GABAergic signaling in the effect of short-term monocular deprivation is directly supported by MR Spectroscopy data in adult humans, showing that resting GABA in a large region of the occipital cortex is specifically reduced after short-term monocular deprivation (*Lunghi et al., 2015b*).

The functional relevance of the BOLD changes we observe is demonstrated by their correlation with our behavioral assay of plasticity, obtained through binocular rivalry. This correlates both with the BOLD ocular dominance change (relative boost/suppression of the deprived/non-deprived eye), and with the BOLD acuity change for the deprived eye (change of spatial frequency tuning, assessed with our pRF-like modeling approach). The correlation holds despite binocular rivalry being restricted to foveal vision, whereas the assessment of BOLD plasticity is pooled across V1 (including the mid-periphery). This implies that the change of binocular rivalry dynamics is a proxy for the more general plasticity effects that involves the whole primary visual cortex. This finding has long reaching implications, as it could validate the use of binocular rivalry as a biomarker of adult cortical plasticity, based on the neural mechanisms revealed by the present 7T fMRI results. Interestingly, the binocular rivalry phenomenon originates in the primary visual cortex – probably at the earliest stages – and is an expression of the dynamics of excitatory transmission and inhibitory feedback (*Tong et al., 2006*); as such it is a measure that could reflect the overall excitation-inhibition ratio (*van Loon et al., 2013*), and its modulation in plasticity (*Lunghi et al., 2015b*; *Maffei and Turrigiano, 2008*).

Our data support the notion that V1 circuitry may be optimized by perceptual experience (*Fiorentini and Berardi, 1980*). They are also consistent with a large perceptual learning literature suggesting that associative cortical areas retain a high degree of flexibility (*Dosher and Lu, 2017*; *Dosher and Lu, 1999*; *Fuchs and Flügge, 2014*; *Harris et al., 2012*; *Kahnt et al., 2011*; *Karni et al., 1995*; *Lewis et al., 2009*; *Shibata et al., 2012*; *Watanabe and Sasaki, 2015*). Although the monocular deprivations effects observed here are more robust in V1, they are reliable in V2 and V3 as well. However, a clear difference emerges between extra-striate visual areas in the ventral and dorsal stream. While ventral area V4 shows a strong deprivation effect, area V3a, located at a similar tier in the dorsal stream, shows no BOLD change after short-term monocular deprivation. V4 is a primary target of the parvocellular system, which is best stimulated by our highest spatial frequency stimulus; V3a and hMT+ are preferential targets of the magnocellular system, which respond more strongly to our lower spatial frequency stimuli (see *Figure 7*). The different plasticity response of the ventral and dorsal stream, together with the selectivity for the high spatial frequencies of the V1 plasticity, suggests that the parvocellular pathway is most strongly affected by short-term plasticity. This is consistent with the finding in non-human primates that deprivation of the stimuli that optimally drive the parvocellular system is sufficient to produce a reliable plasticity effect (*Begum and Tso, 2016*). It is also consistent with the finding that the effect of short-term monocular deprivation is strongest and more long-lasting for chromatic equiluminant stimuli (*Lunghi et al., 2013*).

Other evidence shows that short-term deprivation may affect other properties of vision. In particular, selective deprivation of orientation (*Zhang et al., 2009*) or spatial frequency (*Zhou et al., 2014*) or color (*Zhou et al., 2017*) or even simply phase scrambling of the image in one eye (*Bai et al., 2017*) may lead to a boost of the deprived signal. These effects have been interpreted as a form of release of inhibition from the adapted signal (*Zhang et al., 2009*) – a concept that is not distant from homeostatic plasticity, where the network aims to keep overall activity constant. The conceptual border between adaptation and plasticity is fuzzy, given that some mechanisms are shared and both effects have the same outcomes. Be it adaptation or plasticity, the monocular deprivation mechanisms are probably cortical and affect mainly the deprived eye. There is evidence that the boost of the deprived eye is also observed when the two eyes receive equally strong stimulation, but perception of one eye stimulus is suppressed experimentally (by the continuous flash suppression technique, *Kim et al., 2017*); this result dismisses the retinal or thalamic contribution to the deprivation effect. Only in rare occasions does adaptation induce effects that last over days (*McCollough, 1965*), yet our recent work shows that deprivation effects of short-term monocular deprivation is retained across 6 hr sleep (*Menicucci et al., 2018*), consistent with plasticity reinforcement

during sleep (*Raven et al., 2018*; *Timofeev and Chauvette, 2017*). Most importantly, in adult amblyopic patients, short-term monocular deprivation is able to induce improvement of visual acuity and stereovision (*Lunghi et al., 2018*) for up to one year. All this evidence supports the concept that homeostatic plasticity in the human adult cortex may be linked with or may promote more stable forms of Hebbian-like plasticity. This may endorse stable changes even in the adult brain, well after the closure of the critical period. Functional changes in associative cortex in adults have been demonstrated by short-term paired TMS studies (*Chao et al., 2015*). Interestingly, the decay of this functional connectivity change has a similar time-course as the monocular deprivation effect, about one hour. Also, Hebbian changes at the single cell level can be observed in V1 of adult anaesthetized cat, following activity pairing over a similar time-scale (from minutes to a few hours) (*Frégnac et al., 1988*). All these results demonstrate that V1 retains potential for Hebbian plasticity outside the critical period – although it may need particular conditions to exploit such potential.

Understanding homeostatic plasticity and its relation to Hebbian plasticity may be fundamental to open the way to new approaches to treat brain dysfunction. Particularly important is Ocular Dominance plasticity in amblyopia (*Webber and Wood, 2005*), a cortical deficit still without cure in adults, although recent advancements in training procedures are opening new hopes (*Levi and Li, 2009*; *Sengpiel, 2014*). Endorsing plasticity may increase the effectiveness of these treatments and preliminary data from our laboratory suggest that monocular deprivation of the amblyopic eye may indeed boost sensitivity of the deprived eye and improve its acuity (*Lunghi et al., 2018*) – just like an acuity change is revealed by the present BOLD measurements in normally sighted participants. Our data demonstrate that two hours of abnormally unbalanced visual experience is sufficient to induce a functional reorganization of cortical circuits, particularly of the parvocellular pathway, leading to an alteration of basic visual perceptual abilities.

## Materials and methods

**Key resources table**

| Reagent type (species) or resource | Designation | Source or reference | Identifiers | Additional information |
|---|---|---|---|---|
| Software, algorithm | Freesurfer v6.0.0 | (*Fischl et al., 2002*) | SCR_001847 | |
| Software, algorithm | SPM | (*Friston, 2007*) | SCR_007037 | |
| Software, algorithm | Brain Voyager | (*Goebel et al., 2006*) | SCR_006660 | |
| Software, algorithm | FSL | (*Jenkinson et al., 2012*) | SCR_002823 | |
| Software, algorithm | MATLAB | MathWorks | SCR_001622 | |
| Software, algorithm | Psych Toolbox | (*Brainard, 1997*) | SCR_002881 | |

### Experimental model and subject details

#### Human subjects

Experimental procedures are in line with the declaration of Helsinki and were approved by the regional ethics committee [Comitato Etico Pediatrico Regionale—Azienda Ospedaliero-Universitaria Meyer—Firenze (FI)] and by the Italian Ministry of Health, under the protocol 'Plasticità e multimodalità delle prime aree visive: studio in risonanza magnetica a campo ultra alto (7T)' version #1 dated 11/11/2015. Written informed consent was obtained from each participant, which included consent to process and preserve the data and publish them in anonymous form.

Twenty healthy volunteers with normal or corrected-to-normal visual acuity were examined (8 females and 12 males, mean age = 27 years).

## Method details

### Experimental design

Each participant underwent two scanning sessions separated by two hours, during which they were subject to the short-term monocular deprivation procedure described below. Just before each scanning section, their binocular rivalry was measured psychophysically. One (male) participant was excluded because of strong eye dominance tested with binocular rivalry before the deprivation. This left 19 participants (8 females and 11 males) whose complete datasets were entered all analyses. Sample size was set to enable testing for correlations between neuroimaging and psychophysical data. Previous work (*Lunghi et al., 2015b*) reveals a correlation between MR spectroscopy data and binocular rivalry measures r = 0.62 (or higher), which implies a minimum of 17 participants to detect a significant correlation at 0.05 significance level, with test power of 80% (*Lachin, 1981*).

### Short-term monocular deprivation

Monocular deprivation was achieved by patching the dominant eye for 2 hr. The operational definition of dominant eye applied to the eye showing the longer phase durations during the baseline binocular rivalry measurements. Like in previous studies (*Binda and Lunghi, 2017*; *Lunghi et al., 2011*; *Lunghi et al., 2013*), we used a translucent eye-patch made of plastic material allowing light to reach the retina (attenuation 0.07 logUnits, at least 3 times smaller than the threshold for discriminating a full-field luminance decrement (*Knau, 2000*) and more than ten times smaller than the minimum photopic luminance decrement required for shifting the spatial (*Van Nes and Bouman, 1967*) or temporal contrast sensitivity function (*Kelly, 1961*). The patch prevents pattern vision, as assessed by the Fourier transform of a natural world image seen through the eye-patch. During the 2 hr of monocular deprivation, observers were either engaged in the retinotopic mapping experiment (about 30', described below) or they were free to read and use a computer.

### Binocular rivalry

Binocular rivalry was measured in two short sessions (each comprising two runs of 3 min each), immediately before the Pre- and Post-deprivation MR sessions, in a quiet space adjacent to the MR control room. Visual stimuli were created in MATLAB running on a laptop (Dell) using PsychToolbox (*Brainard, 1997*), and displayed on a 15- inch monitor (BenQ). Like in (*Lunghi et al., 2015b*), observers viewed the visual stimuli presented on the monitor at a distance of 57 cm through anaglyph red-blue goggles (right lens blue, left lens red). Responses were recorded with the computer keyboard by continuous alternate keypresses. Visual stimuli were two oblique orthogonal red and blue gratings (orientation:±45°, size: 3°, spatial frequency: 2 cpd, contrast 50%), surrounded by a white smoothed circle, presented on a black uniform background in central vision. Peak luminance of the red grating was reduced to match the peak luminance of the blue one using photometric measures. All included participants had typical binocular rivalry dynamics, with low percentage of mixed percepts (reported for 8.5 ± 2.04% of time on average). Only one participant experienced of mixed percepts for more than 20% of time (exactly for 31.2%) and his data are in line with the others'.

### Stimuli for fMRI

Visual stimuli were projected with an MR-compatible goggle set (VisuaStimDigital, Resonance Technologies, Los Angeles, USA), connected to a computer placed in the control room. The goggles covered a visual field of approximately 32 × 24 deg, with a resolution of 800 × 600 pixels, mean luminance 25 cd/m$^2$; the images in the two eyes were controlled independently.

During all functional MRI scans participants were instructed to maintain central fixation on a red point (0.5 degrees) that was constantly visible at the center of the screen. Bandpass noise stimuli were white noise images filtered to match the spatial frequency tuning of neurons in the visual cortex (*Morrone and Burr, 1988*). We generated a large white noise matrix (8000 × 6000) and filtered it with a two-dimensional circular bandpass filter *Bp* defined by *Equation 1*:

$$Bp = e^{\frac{-\ln\left(\frac{y}{P}\right)^2}{2[q*ln(2)]^2}}$$ (1)

where *P* is the peak spatial frequency, *q* is the filter half-width at half maximum in octaves. We generated five band-pass noise stimuli, by setting q = 1.25 octaves and p = 0.1 cpd, 0.2 cpd, 0.4 cpd,

1.1 cpd, 2.7 cpd. Each stimulus was presented for a block of 3 TRs, during which the image was refreshed at 8 Hz (randomly resampling a 800 × 600 window from the original matrix). Stimuli were scaled to exploit the luminance range of the display, and this yielded very similar RMS contrast values (shown in *Figure 4—figure supplement 1*). Stimulus blocks were separated by 4 TRs blanks, consisting of a mid-level gray screen. The five band-pass noise stimuli blocks were presented in pseudo-random order, twice per run, for a total of 70 TRs. In each run, stimuli were only presented to one eye, while the other was shown a mid-level gray screen. Each eye was tested once, before and after deprivation.

Immediately upon application of the monocular patch, we performed two additional scans to perform retinotopic mapping of visual areas. Meridian and ring stimuli were presented monocularly (to the non-patched eye) and were defined as apertures of a mid-level gray mask that uncovered a checkerboard pattern, 1 deg at 1 deg eccentricity to 2.5 deg at 9 deg eccentricity, rotating and contracting at a rate of one check per second. Meridians were defined by two 45° wedges centered around 0° or around 90°. The horizontal and vertical meridian were presented interchangeably for 5 TRs each (without blanks) and the sequence was repeated six times for a total of 60 TRs. Rings partitioned screen space into six contiguous eccentricity bands (0–0.9 deg, 0.9–1.8 deg, 1.8–3.3 deg, 3.3–4.7 deg, 4.7–6.48 deg, 6.48–9 deg). Odd and even rings were presented in two separate runs. In each run, the three selected rings and one blank were presented in random order for 5 TRs each, and the sequence was repeated (with different order) 6 times for a total of 120 TRs.

## MR system and sequences

Scanning was performed on a Discovery MR950 7 T whole body MRI system (GE Healthcare, Milwaukee, WI, USA) equipped with a 2-channel transmit driven in quadrature mode, a 32-channel receive coil (Nova Medical, Wilmington, MA, USA) and a high-performance gradient system (50 mT/m maximum amplitude and 200 mT/m/ms slew rate).

Anatomical images were acquired at 1 mm isotropic resolution using a T1-weighted magnetization-prepared fast Fast Spoiled Gradient Echo (FSPGR) with the following parameters: TR = 6 ms, TE = 2.2 ms. FA = 12 deg, rBW = 50 kHz, TI = 450 ms, ASSET = 2.

Functional images were acquired with spatial resolution 1.5 mm and slice thickness 1.4 mm with slice spacing = 0.1 mm, TR = 3 s, TE = 23 ms, rBW = 250 kHz, ASSET = 2, phase encoding direction AP-PA. No resampling was performed during the reconstruction. For each EPI sequence, we acquired two additional volumes with the reversed phase encoding direction.

## Quantification and statistical analysis
### ROI definition

Areas V1, V2 and V3 were manually outlined for all participants using retinotopic data projected on surface models of white matter. The V1/V2 boundary was traced from the vertical/horizontal meridian flip superior/inferior to the calcarine sulcus, and the V2/V3 border and V3 border from the subsequent opposite flips. Areas V4, V3a and hMT+ (merging MT and MST) were defined based on the cortical parcellation atlas by Glasser et al. (*Glasser et al., 2016*). V1, V2, V3, V4 and V3a ROIs were further restricted to select the representation of our screen space. Specifically, the anterior boundaries were defined based on activation from most peripheral (6.48°−9°) ring stimuli of the retinotopic mapping scans; in addition, vertices were only included in the analysis if their preferred eccentricity (estimated through Population Receptive Field modelling, see below) was larger than 1, since no reliable mapping could be obtained for the central-most part of the visual field.

## Pre-processing of imaging data

Analyses were performed mainly with Freesurfer v6.0.0, with some contributions of the SPM12 and BrainVoyager 20.6 and FSL version 5.0.10 (*Jenkinson et al., 2012*) packages.

Anatomical images were corrected for intensity bias using SPM12 (*Friston, 2007*) and processed by a standard procedure for segmentation implemented in Freesurfer (recon-all: *Fischl et al., 2002*). In addition, each hemisphere was aligned to a left/right symmetric template hemisphere (fsaverage_-sym: *Greve et al., 2013*).

Functional images were corrected for subject movements (*Goebel et al., 2006*) and undistorted using EPI images with reversed phase encoding direction (Brain Voyager COPE plug-in *Jezzard and*

*Balaban, 1995*). We then exported the preprocessed images from BrainVoyager to NiFTi format. These were aligned to each participant's anatomical image using a boundary based registration algorithm (Freesurfer *bbergister* function) and projected to the cortical surface of each hemisphere. All analyses were conducted on data in the individual subject space. In addition, for visualization purposes, we also aligned the results of timecourse analyses (GLM and subsequent pRF and spatial frequency tuning estimates) to the left/right symmetric template hemisphere. Averaged results across the 18 × 2 hemispheres are shown in the maps of *Figure 1B*, *Figure 5A* and *Figure 1—figure supplement 1*.

## GLM analysis of fMRI data

General Linear Model analysis was performed with in-house MATLAB software (*D'Souza et al., 2016*). We assumed that fMRI timecourses result from the linear combination of N predictors: boxcar functions representing stimulus presence/absence (one per stimulus type) convolved by a standard hemodynamic response function (see *Equation 2*), plus two nuisance variables (a linear trend and a constant). We modeled the hemodynamic response function as a gamma function h(t):

$$h(t) = \frac{\left(\frac{t-\delta}{\tau}\right)^{(n-1)} e^{-\left(\frac{t-\delta}{\tau}\right)}}{\tau(n-1)!} \tag{2}$$

with parameters *n* = 3, *t* = 1.5 s, and d = 2.25 s (*Boynton et al., 1996*). Beta weights of the stimuli predictors were taken as estimates of the BOLD response amplitude and normalized by the predictor amplitude to obtain a measure that directly corresponds to % signal change; beta weights were also scaled by an error measure to obtain t-values, following the same procedure as in (*Friston et al., 1994*). Computing BOLD responses for each individual vertex of the cortical surface leads to up-sampling the functional data (each 1.5 × 1.5 × 1.5 mm functional voxel projecting on an average of 3 vertices). We ensured that this does not affect our statistical analyses by first averaging data from all vertices within a region of interest (e.g. V1), thereby entering all ANOVAs with a single value per subject and region of interest.

## Population receptive field mapping

The pRFs of the selected voxels were estimated with custom software in MATLAB, implementing a method related to that described by Dumoulin and Wandell (Dumoulin & Wandell, 2008) and provided as supplementary material. We modeled the pRF with a 1D Gaussian function defined over eccentricity, with parameters $\varepsilon$ and $\sigma$ as mean and standard deviation respectively, and representing the aggregate receptive field of all neurons imaged within the vertex area. We defined the stimulus as a binary matrix S representing the presence of visual stimulation over space (here, eccentricity between 0 and 10 deg with 40 steps per deg) for each of 6 ring stimuli. We used the results of our GLM analysis to estimate the vertex response to each of our 6 rings (as t-values; using beta values yields very similar results). We assumed that each vertex response is the linear sum over space (eccentricity) of the overlap between the pRF of the voxel and the input stimulus, which is mathematically equivalent to the matrix multiplication between the stimulus and the pRF.

$$R_i = G(\varepsilon, \sigma) * S_i \tag{3}$$

where *i* is the index to ring number and varies between 1 and 6.

We used this equation to predict the response to our six rings for a large set of initial pRF parameters; for each vertex, we measured the correlation (our goodness-of-fit index) between the predicted response and the observed t-values. If the highest correlation was .7 the vertex was discarded; otherwise, the parameters yielding the highest correlation were used to initialize a nonlinear search procedure (MATLAB simplex algorithm), which manipulated $\varepsilon$ and $\sigma$ to maximize goodness-of-fit, with the constraint that $\varepsilon$ could not exceed 20 deg or be smaller than 1 deg, and $\sigma$ could not be smaller than. 1 deg. Successful fits were obtained for 72.00 ± 1.86% of V1 vertices, for which the initial coarse grid search gave a correlation > 0.7 and the nonlinear search settled within the constraints. All analyses (on average and distribution of responses and tuning parameters) considered the sub-region of V1 for which a successful fit was obtained. We used $\varepsilon$ to estimate the preferred eccentricity of each vertex.

The main modifications of our procedure relative to that described by Dumoulin and Wandell (*Dumoulin and Wandell, 2008*) are the following: (a) fMRI data were acquired in a block design with only six stimulus types (six eccentricity bands) rather than varying stimulus position at each TR; this allowed us to use a standard GLM approach to estimate each vertex response to the six stimuli (assuming a standard hemodynamic response function) and then use the pRF model to predict these six time-points – much faster than predicting the full fMRI series of 120 × 2 TRs; (b) our stimuli and consequently our pRFs were defined in one dimension (eccentricity) – whereas the standard pRF is defined in 2D, eccentricity and polar angle (or Cartesian x and y); (c) we maximized the correlation between the predicted and observed fMRI response time-courses rather than minimizing the root mean square error; this eliminates the need to estimate a scale factor to account for the unknown units of the BOLD signal.

## Population tuning for spatial frequency

Using a similar logic, we also estimated the population tuning for Spatial Frequency, which represents the aggregate Spatial Frequency tuning of the population of neurons imaged within each vertex area. We modeled the population tuning using a family of functions that includes the psychophysical Contrast Sensitivity Function (CSF) and can be specified by the following one-parameter equation (Difference-of-Gaussians):

$$pSFT = e^{\frac{-v^2}{\sigma}} - - \; e^{\frac{-v^2}{\sigma/50}} \times \sigma \tag{4}$$

Like we did for the pRF mapping, we defined a stimulus matrix S representing the Fourier spectra of our five bandpass noise stimuli, that is the energy of visual stimulation in the frequency domain (here, between 0.03 cpd and 12.5 cpd) for each stimulus. We used the results of our GLM analysis to estimate the vertex response to each of our five bandpass noise stimuli (as t-values; using beta values yields very similar results). We assumed that each vertex response is the linear sum over frequency of the overlap between the pSFT of the voxel and the input stimulus, which is mathematically equivalent to the matrix multiplication between the stimulus and the pSFT.

Like for pRFs, we estimated the best-fit $\sigma$ parameter of each vertex pSFT with a two-step procedure: a coarse-grid search followed by the simplex search. We used the matrix multiplication of the pSFT and the stimulus to predict the response to our five bandpass noise stimuli for a large set of initial $\sigma$ values (between 1 and 1,000 in 100 logarithmic steps); for each vertex, we measured the correlation (our goodness-of-fit index) between the predicted response and the observed t-values. If the highest correlation was <.5, the voxel was discarded, otherwise the parameter yielding the highest correlation were used to initialize a nonlinear search procedure (MATLAB simplex algorithm), which manipulated $\sigma$ to maximize goodness-of-fit, with the constraint that $\sigma$ could not be smaller than. 3 and larger than 10,000. Successful fits were obtained for 88.84 ± 1.28% of V1 vertices for which we obtained a successful eccentricity fit (86.77 ± 1.25% of all V1 vertices).

We express the $\sigma$ parameter in terms of the high-spatial frequency cutoff of the filter (highest spatial frequency at half maximum), $SFco$ for each vertex:

$$\sqrt{\frac{\sigma}{2}} \tag{5}$$

## Indices defining the effect of deprivation

We computed the effects of short-term monocular deprivation on both the dynamics of binocular rivalry and our fMRI results, estimating the degree to which the two measures are correlated. In all cases, the same equation was applied to psychophysical and fMRI data.

The first index, called 'Deprivation Index' or $DI_{psycho}$ and $DI_{BOLD}$ is given by *Equation 6*

$$DI = \left(\frac{y_{DepPOST}}{y_{DepPRE}}\right) / \left(\frac{y_{NdepPOST}}{y_{NdepPRE}}\right) \tag{6}$$

For psychophysics, y = mean duration of Binocular Rivalry phases of the Dep or Ndep eye, during the PRE- or POST deprivation sessions; for fMRI, y = mean BOLD response across V1 vertices to stimuli in the Dep or Ndep eye, during the PRE- or POST-deprivation sessions.

The second index, called 'Deprived-eye change' or $DepC_{psycho}$ and $DepC_{cutoff}$ is given by *Equation 7*

$$DepC = \left(\frac{y_{DepPOST}}{y_{DepPRE}}\right) \tag{7}$$

For psychophysics, y = mean duration of Binocular Rivalry phases of the Dep eye, during the PRE- or POST deprivation sessions. For fMRI, y = mean spatial frequency cut-off across V1 vertices estimated for stimuli in the Dep eye, during the PRE- or POST-deprivation sessions.

## Statistics

Data from individual participants (mean binocular rivalry phase durations or mean BOLD responses/ pRF/pST across V1 or V2 vertices) were analyzed with a repeated measure ANOVA approach, after checking that distributions do not systematically deviate from normality by means of the Jarque-Bera test for composite normality (MATLAB *jbtest* function, p-values given in the relevant figures). F statistics are reported with associated degrees of freedom and p-values in the Results section, in the form: $F(df, df_{err}) = value$; p = value. Post-hoc paired t-tests comparing conditions follow the ANOVA results, in the form: t(df)= value, p = value. Associations between variables are assessed with Pearson product-moment correlation coefficient, reported in the form: r(n) = value, p = value. Aggregate subject data (i.e. vertices pooled across participants and hemispheres) were typically non-normally distributed and thereby were analysed with non-parametric tests. The Wilcoxon sign-rank test was used for comparing medians, and results are reported in the form: z = value, p = value.

## Acknowledgments

This research was supported by the European Research Council under the European Union's Seventh Framework Programme (FPT/2007–2013) under grant agreement number 338866 (PB, CL, and MCM) and under ERA-NET project 'Neuro-DREAM' (CL, and MCM) and by the European Union's Horizon 2020 Research and Innovation Programme under the Marie Sklodowska-Curie grant agreement number 641805 (JWK) and by the Italian Ministry of University and Research under the project PRIN2015 (MCM) and by Fondazione Roma under the Grants for Biomedical Research: Retinitis Pigmentosa (RP)-Call for proposals 2013- 'Cortical Plasticity in Retinitis Pigmentosa: an Integrated Study from Animal Models to Humans'. The authors would like to thank Mauro Costagli for help with the data acquisition, and David C Burr for his comments on the manuscript.

## Additional information

### Funding

| Funder | Grant reference number | Author |
| --- | --- | --- |
| European Research Council | ERC ECSPLAIN 338866 | Paola Binda<br>Jan W Kurzawski<br>Maria Concetta Morrone |
| H2020 European Institute of Innovation and Technology | NextGenVis 641805 | Jan W Kurzawski |
| Ministero dell'Istruzione, dell'Università e della Ricerca | PRIN2015 | Claudia Lunghi<br>Maria Concetta Morrone |
| European Research Council | ERA-NET Neuro-DREAM | Claudia Lunghi<br>Maria Concetta Morrone |
| Fondazione Roma | | Maria Concetta Morrone |

The funders had no role in study design, data collection and interpretation, or the decision to submit the work for publication.

## Author contributions
Paola Binda, Conceptualization, Data curation, Software, Formal analysis, Investigation, Methodology, Writing—original draft, Writing—review and editing; Jan W Kurzawski, Data curation, Software, Formal analysis, Investigation, Writing—review and editing; Claudia Lunghi, Conceptualization, Investigation, Methodology, Writing—review and editing; Laura Biagi, Data curation, Investigation; Michela Tosetti, Resources, Supervision; Maria Concetta Morrone, Conceptualization, Supervision, Funding acquisition, Writing—original draft, Writing—review and editing

## Author ORCIDs
Paola Binda http://orcid.org/0000-0002-7200-353X
Jan W Kurzawski http://orcid.org/0000-0003-2781-1236
Claudia Lunghi http://orcid.org/0000-0003-3811-5404
Laura Biagi http://orcid.org/0000-0003-2159-439X
Michela Tosetti http://orcid.org/0000-0002-2515-7560
Maria Concetta Morrone http://orcid.org/0000-0002-1025-0316

## Ethics
Human subjects: Experimental procedures are in line with the declaration of Helsinki and were approved by the regional ethics committee [Comitato Etico Pediatrico Regionale-Azienda Ospedaliero-Universitaria Meyer-Firenze (FI)] and by the Italian Ministry of Health, under the protocol 'Plasticità e multimodalità delle prime aree visive: studio in risonanza magnetica a campo ultra alto (7T)'.

## Decision letter and Author response
Decision letter https://doi.org/10.7554/eLife.40014.019
Author response https://doi.org/10.7554/eLife.40014.020

# Additional files

## Supplementary files
• Transparent reporting form
DOI: https://doi.org/10.7554/eLife.40014.015

## Data availability
BOLD responses and pRF fits as shown in all figures (main and supplementary) have been deposited on Dryad, through a link provided with the current submission (doi:10.5061/dryad.tp24j18). Custom Matlab code, used for pRF fitting, is included as Source code file 1.

The following dataset was generated:

| Author(s) | Year | Dataset title | Dataset URL | Database and Identifier |
|---|---|---|---|---|
| Binda P, Kurzawski JW, Lunghi C, Biagi L, Tosetti M, Morrone MC | 2018 | Short-term plasticity of the human adult visual cortex measured with 7T BOLD | https://dx.doi.org/10.5061/dryad.tp24j18 | Dryad Digital Repository, 10.5061/dryad.tp24j18 |

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
