## [Decision Letter]

Thank you for submitting your article "Short-term plasticity of the human adult visual cortex measured with 7T BOLD" for consideration by *eLife*. Your article has been reviewed by two peer reviewers, and the evaluation has been overseen by a Reviewing Editor and Joshua Gold as the Senior Editor. The reviewers have opted to remain anonymous.

The reviewers have discussed the reviews with one another and the Reviewing Editor has drafted this decision to help you prepare a revised submission.

Summary:

The paper examined the consequences of depriving one eye of pattern vision for two hours in adult humans, as assessed with functional MRI and binocular rivalry. The measurements were made both immediately before and after the 2-hour deprivation period. Among the major findings are altered BOLD response which increased for stimuli viewed through the deprived eye and decreased for stimuli viewed through the non-deprived eye, particularly for stimuli comprised of high spatial frequencies. The deprived eye was also favored during binocular rivalry. These effects were correlated across participants: those who showed the biggest BOLD effect also showed the biggest rivalry effect. The interpretation is that visual cortex retains a high degree of plasticity in adult humans, especially for signals carried by the parvocellular pathways.

The manuscript was well received by both reviewers. They were both positive about the approach, the strength of the data and its analysis as well as on the clarity of the presentation. However, there was some significant concerns about the interpretation, and the reviewers highlighted several problems that must be addressed before the paper will be considered for publication in *eLife*. These are listed below.

Essential revisions:

1) Both reviewers were concerned with the way the problem addressed in the paper was framed and the results interpreted. They pointed out that – contrary to the message conveyed by the paper that the adult cortex is hard-wired – short-term plasticity such as adaptation in the adult visual cortex is well known and not particularly controversial. This issue should be addressed with appropriate references to the existing literature. This point was initially raised by reviewer 1 and was shared by the reviewer 2 during the discussion that followed.

2) Please clarify how (and whether) your data showing short-term plasticity in the adult brain relate to the critical periods and long-term effects of deprivation during development. This is of particular interest because of the opposite direction of the short-term effects in the adult and the effects of deprivation during development.

3) Please explain your interpretation of ocular dominance columns and ocular preferences.

4) If possible, provide fMRI maps of responses to stimuli presented to each eye. Of particular interest are potential differences in the effects for central and peripheral visual fields that could emerge from such a comparison.

5) Please discuss how the removal of phase information during deprivation may have affected your results.

6) Please address the question whether stimulation of the nondeprived eye during retinotopic mapping during deprivation may have affected the outcome of your study.

7) Please address the reliability of the analysis of MT activation. The small number of voxels could make it difficult to assess the result.

8) Could you relate the results to studies of neural adaptation to deprivation in keratoconus, a relatively common condition that affects young adults? This work should be referenced here.

9) The phrase '… sensory cortex is resilient to plastic change' is used multiple times including the Abstract and Discussion. We think 'resilient' should be 'resistant', and 'plastic change' is redundant, and should perhaps just be 'change'. We suspect the authors have an idea of 'plastic change' that is different from other kinds of change, but if so they should define it.

*Reviewer #1:*

Binda and colleagues examined the consequences of depriving one eye of pattern vision for 2 hours in adult humans, as assessed with functional MRI and binocular rivalry. The measurements were made both immediately before and after the 2-hour deprivation period, with a few main findings. First, the BOLD response in visual cortex was altered by the deprivation, increasing for stimuli viewed through the deprived eye and decreasing for stimuli through the non-deprived eye, particularly for stimuli comprised of high spatial frequencies. Second, binocular rivalry tended to favor the deprived eye. Third, the two effects were correlated across participants: those who showed the biggest BOLD effect also tend to show the biggest rivalry effect. The interpretation is that visual cortex retains a high degree of plasticity in adult humans, especially for signals carried by the parvocellular pathways.

The study is well done and the results are clear. The correlation between the BOLD effects and psychophysics at the level of individual subjects is an important observation. The quantification of the results as a function of stimulus spatial frequency content and visual area is useful and leads to several interesting patterns of results. For all of these reasons, this is a generally strong study.

However, there is one major problem with the paper as written (as I see it), and that is the framing and interpretation. A reader might come away with the mistaken impression that some scientists think adult cortex has no plasticity, whereas others think it does, and that this study will resolve the issue. For example, the authors cite papers in support of the claims that adult cortex is essentially hard-wired (Mitchell and Sengpiel, 2009; Sato and Stryker, 2008) or is characterized more by stability than plasticity (Baseler et al., 2002; Baseler et al., 2011; Wandell and Smirnakis, 2009). As far as I can tell, none of these papers argues against the kind of short-term, adaptation effects observed here. In fact, the Wandell and Smirnakis paper goes so far as to state, 'There can be no serious debate as to whether the brain is plastic or not: it is both.'

While there is no hard-line separating adaptation from plasticity, this paper is clearly framed in terms of plasticity, based on both the title and the motivation. For example, the Introduction discusses the issue of critical periods. Yet it is not clear what this paper has to do with critical periods, or with the kind of long-term effects seen with deprivation in early development, since all of the effects in this study are measured within minutes of the deprivation (which some might call contrast adaptation). As the authors note in the Discussion, the effects are in the opposite direction of deprivation effects in development. In development, deprivation results in long-term loss of function. Here, as in many adaptation papers, the deprivation results in a short-term boost in function.

Such short-term effects of adaptation of the early visual system are widely known in vision science. For example, twenty minutes in the dark will greatly change psychophysical sensitivity and presumably BOLD responses to faint flashes of light. Few would argue that this demonstrates that adult cortex remains plastic in the same way as in early development. Similarly, Mon-Williams and colleagues (1998) showed that 30 minutes of defocus caused increased acuity and a change in spatial frequency sensitivity. There are also studies showing the effects of contrast adaptation on BOLD responses (e.g. Gardner et al., 2005).

In short, the notion that early visual cortex in adults alters its response properties following prolonged exposure to particular stimuli is widely known and uncontroversial. The results do not support the authors' main conclusion: 'Given these strong indications that the effect of monocular deprivation is a form of plasticity, our observation that the effect is present in V1 calls for a reconsideration of the established idea that human adult primary sensory cortex is resilient to plastic change.' Most of the evidence cited in the Discussion that these effects reflect substantial, long-term changes in the visual system, akin to developmental plasticity, is based on other work, not the experiments in this paper.

Nonetheless, as indicated above, the study is a very nice characterization of visual system responses to monocular contrast adaptation. The fMRI results are interesting and well characterized, and the relationship between the BOLD changes and the behavioral changes make for a nice contribution.

Finally, there is one issue with regard to the BOLD results that I found puzzling. The authors measure the ocular dominance of cortical locations ('surface vertices') but do not want the reader to confuse this with the notion of ocular dominance columns. What, exactly, is the interpretation of reasonably reliable and repeatable measures of ocular preference?

*Reviewer #2:*

This is a fascinating, well-designed and technically-sophisticated study demonstrating short-term visual plasticity in adult human visual cortex. In particular, the authors show evidence that just 2 hours of monocular deprivation induces changes in 7T BOLD signal related to ocular dominance, which correlate strongly with behavioral performance (binocular rivalry) in the same subjects. The methods used to analyze the fMRI results include several thoughtful determinations, including the extraction (and subsequent assessment of change in) population tuning for spatial frequencies, and the ability to assess that changes in BOLD signal post-deprivation occurred preferentially in units that were selective for the non-deprived eye prior to the onset of deprivation.

I have some suggestions for additional information and discussion I would like to see included:

1) The translucent eye patch used to induce the 2 hours of visual deprivation allowed light to reach the retina, but prevented pattern vision. This patch did not only remove high spatial frequency and orientation information (I assume) but likely also removed phase information from the scene. Is this correct and/or relevant to the present experiments and findings? If so, can the authors discuss what implications this may have had in terms of impact on eye specific processing and brain activity measured presently.

2) In the Materials and methods, it becomes clear that the subjects were not just idly and freely reading or using a computer during the 2 hours of deprivation, as stated in the subsection “Short-term Monocular Deprivation”, but that they underwent retinotopic fMRI "immediately after application of the patch". What proportion of the 2 hours was spent with the non-deprived eye being stimulated with the high-contrast, broadband stimuli typical of retinotopic mapping? Given the clear demonstration that only 2 hours of monocular form deprivation appears to have significant impact on neural properties in the visual system, could the stimulation received by the non-deprived eye have influenced the outcome obtained here and if so, how?

3) In the fMRI flat maps shown (Figures 1B, 5A), area MT appears to have *very* few significant voxels. So few in fact, that one wonders how reliable the analyses performed with respect to MT are. Please discuss.

4) Are the authors certain that the area they label as MT is truly MT, and not the MT complex, or hMT+?

5) In the Results, subsection “Figure 4: Deprivation affects spatial frequency selectivity in V1”, it is claimed that prior to deprivation, the BOLD response in V1 has broad-band selectivity with peak SF around 1 cpd. However, the graph in Figure 4A suggests that the peak is closer to 0.4 to just under 1 cpd. Please fix or discuss.

6) In the context of both motivating the present study (i.e. in the Introduction) and providing an example of how the present knowledge gained could be applied in the context of a clinical population, reference should be made to keratoconic patients. Much more so than amblyopia, keratoconus develops in early adulthood, well after the end of the critical period for ocular dominance plasticity. Moreover, just like the deprivation used here, it does not deprive the eye of light, but rather introduces severe optical aberrations, largely to one eye in many cases, filtering out high spatial frequency and phase information. Several excellent studies of neural adaptation to keratoconus have been published (see Sabesan, Yoon and colleagues, among others), at least some of which should be referenced in the present work.

---

## [Author Response]

Essential revisions:1) Both reviewers were concerned with the way the problem addressed in the paper was framed and the results interpreted. They pointed out that – contrary to the message conveyed by the paper that the adult cortex is hard-wired – short-term plasticity such as adaptation in the adult visual cortex is well known and not particularly controversial. This issue should be addressed with appropriate references to the existing literature. This point was initially raised by reviewer 1 and was shared by the reviewer 2 during the discussion that followed.

We accept the criticism and now discuss the literature that demonstrates how the adult brain does indeed retain the ability to change in response to experience. We also changed the title and most of the Introduction. We share the view that the border between adaptation and adult plasticity is not a sharp one and that we cannot definitely assign the effects of short-term monocular deprivation to either phenomenon (plasticity or adaptation). Lacking direct evidence that short-term deprivation in adults leads to anatomical reorganization of the cortex, we decided to stress the functional aspect. We have consequently rephrased the title (from “Short-term plasticity of…” to “Response to short-term deprivation…”) and made this one of the key points of the Introduction. We are particularly grateful for the suggestion to discuss the literature on short- and long-term effects of blur and higher order aberrations (including keratoconus). Optical defocus causes both enhancement (especially in the short-term), and suppression (if maintained for a long period or a lifetime), suggesting that the two phenomena are linked and that, in essence, the results of optical defocus parallel the effects of monocular deprivation that we measured.

2) Please clarify how (and whether) your data showing short-term plasticity in the adult brain relate to the critical periods and long-term effects of deprivation during development. This is of particular interest because of the opposite direction of the short-term effects in the adult and the effects of deprivation during development.

This important point is now directly addressed in the Introduction.We agree that the link between the short-term effects in the adult and the long-term effects seen in development is largely unexplored. However, there is animal work indicating that multiple forms of plasticity co-exist in the visual cortex: Hebbian plasticity (leading to suppression of the deprived input through a change of synaptic connectivity) and homeostatic plasticity (leading to the opposite effect, boosting the deprived input, through response gain modulation). Turrigiano and coworkers (but also many others) suggest that the two phenomena correspond to two sides of the same coin, with homeostatic plasticity acting to stabilize the network after the potentially destabilizing effects of Hebbian plasticity (e.g. LTP implements a positive feedback loop, bound to lead to excessive excitation in the absence of a mitigating mechanism). This opens the possibility that stimulating homeostatic plasticity (possibly of the form we observe here) might be instrumental to reinstating Hebbian plasticity in the adult, which could explain the efficacy of the short-term deprivation paradigm in improving acuity in adult amblyopes. However, irrespectively of the interpretation, it is a fact that a boost of the deprived information has been repeatedly observed, both during critical period and in adulthood, in mice and rat visual cortex (thoroughly reviewed in Maffei and Turrigiano, 2008; Turrigiano, 2012; Turrigiano and Nelson, 2004).

3) Please explain your interpretation of ocular dominance columns and ocular preferences.

Our voxel resolution does not allow to detect individual ODC or the “stripes” they form on the cortical surface. However we were forced to use large voxels size of 1.5mm to increase s/n ratio of the acquisition, given the limited time to measure the effect at its maximum. Also the positioning of the acquisition slices was not optimized for ODC imaging – slice prescription being aimed at covering both ventral and dorsal visual areas. Due to these limitations of our technique, we cannot directly image ODCs and how monocular deprivation affects them. However, each of our voxels draws a biased sample of neurons spanning several ODCs, and this bias leads to the eye preference of that particular voxel. We stress the need for further studies that reach a higher level of anatomical detail and might actually image a dynamic change of the ODC – if it occurs. Text clarifying this point has been added in the Results and Discussion.

4) If possible, provide fMRI maps of responses to stimuli presented to each eye. Of particular interest are potential differences in the effects for central and peripheral visual fields that could emerge from such a comparison.

Two new panels in Figure 1—figure supplement 1 show maps of average BOLD responses (% signal change in response to the highest SF stimulus) separately for the two eyes, before and after deprivation. The maps support the main analyses showing that the two eyes undergo opposite effects: suppression for the non-deprived, enhancement for the deprived. In addition, the maps show that the effects are spread across the V1 territory (they do not cluster in the fovea/periphery). This homogenous retinotopic distribution of the effects is consistent with the homogeneous change of spatial frequency tuning observed across eccentricity (Figure 5).

5) Please discuss how the removal of phase information during deprivation may have affected your results.

We apologize for the lack of clarity on this methodological aspect. Contrary to what was done in Lunghi et al. (2016), here we used a monocular patch that removed all contrast at a large range of spatial frequencies: leaving no power, hence no phase information – although the patch would of course preserve temporal luminance contrast (abrupt change of overall luminance) without phase alteration. Thus, our data cannot speak to the issue of amplitude attenuation or phase scramble as deprivation signal; however, it is interesting to note that the mere removal of phase information, with no manipulation of contrast, is sufficient to induce a short-term deprivation effect (Bai et al., 2017). This issue is now addressed in the Discussion section.

6) Please address the question whether stimulation of the nondeprived eye during retinotopic mapping during deprivation may have affected the outcome of your study.

Retinotopic mapping was performed over a relatively small portion of the deprivation time (less than 30’), and we have revised the text to state this more clearly. We agree that the kind of stimulation provided during the retinotopic mapping experiment is highly atypical and could have affected the outcome of the monocular deprivation procedure. In principle, increasing contrast in the non-deprived eye should have the same consequences as decreasing contrast in the deprived eye (through the patch). If so, the two manipulations could add up, implying that the MD protocol used in this experiment produces exaggerated consequences (and possibly larger V1 changes) compared to the previously published MD effects. However, this is not the case. From a re-analysis of recent published studies on MD and binocular rivalry in human adults, we find that the average effect size is 1.78 ± 0.20 (measured as the deprived/non-deprived ratio after deprivation on 46 subjects) and that the current effect (average ratio = 1.71 ± 0.32 on 19 subject) is well within this “normal” range. This argument is now reported in the Results section (near the description of Figure 3).

7) Please address the reliability of the analysis of MT activation. The small number of voxels could make it difficult to assess the result.

We acknowledge and clarify this issue, which we discuss in the Results section (near the description of Figure 7, edited to include an additional panel with hMT+ average BOLD responses). Significance of the activation map of Figure 1 is calculated vertex by vertex and FDR corrected; with this strict test, activation maps indeed fail to reveal a significant response in hMT+. We believe that this is due to increased variability of BOLD responses in individual vertices within this region of interest, compared to the other regions. This could be due to more variable retinotopy in hMT+ compared to the other regions (Smith et al., 2006) and/or more variable mapping of spatial frequencies. To factor out this variability, and make a fair comparison across regions, we average activity across all vertices within the ROI. The result of this analyses is presented in Figure 7, where the SF response curve is shown for V4, V3A and now also for hMT+. Once vertices within the ROI are pooled, hMT+ and V3A behave similarly, showing a low-pass behavior and yet a strong and significant response to the highest SF stimulus. Importantly both ROIs display no effect of monocular deprivation. Not only the deprivation index (bottom panel) in hMT+ (and V3a) is non-significantly different from 0, which could either mean no effect of deprivation or too variable/unreliable effects to reach statistical significance, it is also significantly smaller than V4, which means that the hMT+ (and V3a) measurements do afford sufficient statistical power to measure a statistically reliable difference.

8) Could you relate the results to studies of neural adaptation to deprivation in keratoconus, a relatively common condition that affects young adults? This work should be referenced here.

We have included reference to this interesting and relevant literature that we had unfortunately overlooked. Keratoconus, and other optical distortions, appear to be accompanied by both enhancement (Sabesan and Yoon, 2010) and suppression (Sabesan and Yoon, 2009), in a sense recapitulating the effects of monocular patching. We have stressed this similarity and added mention to this condition as further motivation to study the mechanisms that allow the visual system to adjust to experience.

9) The phrase '… sensory cortex is resilient to plastic change' is used multiple times including the Abstract and Discussion. We think 'resilient' should be 'resistant', and 'plastic change' is redundant, and should perhaps just be 'change'. We suspect the authors have an idea of 'plastic change' that is different from other kinds of change, but if so they should define it.

Thank you for pointing this out. We eliminated repetitions and the term resilient. We have edited the text to introduce the term “functional plasticity” (referring to the effects of short-term deprivation and related phenomena in the adult brain) and differentiated it from “structural changes”, which we use with reference to the plasticity phenomena typically triggered within the critical period.

Reviewer #1:[…] However, there is one major problem with the paper as written (as I see it), and that is the framing and interpretation. A reader might come away with the mistaken impression that some scientists think adult cortex has no plasticity, whereas others think it does, and that this study will resolve the issue. For example, the authors cite papers in support of the claims that adult cortex is essentially hard-wired (Mitchell and Sengpiel, 2009; Sato and Stryker, 2008) or is characterized more by stability than plasticity (Baseler et al., 2002; Baseler et al., 2011; Wandell and Smirnakis, 2009). As far as I can tell, none of these papers argues against the kind of short-term, adaptation effects observed here. In fact, the Wandell and Smirnakis paper goes so far as to state, 'There can be no serious debate as to whether the brain is plastic or not: it is both.'

We acknowledge and clarify this issue (see point 1 and 2 of the reply to the editor, also reported below) and thank the reviewer for this comment.

While there is no hard-line separating adaptation from plasticity, this paper is clearly framed in terms of plasticity, based on both the title and the motivation. For example, the Introduction discusses the issue of critical periods. Yet it is not clear what this paper has to do with critical periods, or with the kind of long-term effects seen with deprivation in early development, since all of the effects in this study are measured within minutes of the deprivation (which some might call contrast adaptation). As the authors note in the Discussion, the effects are in the opposite direction of deprivation effects in development. In development, deprivation results in long-term loss of function. Here, as in many adaptation papers, the deprivation results in a short-term boost in function.Such short-term effects of adaptation of the early visual system are widely known in vision science. For example, twenty minutes in the dark will greatly change psychophysical sensitivity and presumably BOLD responses to faint flashes of light. Few would argue that this demonstrates that adult cortex remains plastic in the same way as in early development. Similarly, Mon-Williams and colleagues (1998) showed that 30 minutes of defocus caused increased acuity and a change in spatial frequency sensitivity. There are also studies showing the effects of contrast adaptation on BOLD responses (e.g. Gardner et al., 2005).In short, the notion that early visual cortex in adults alters its response properties following prolonged exposure to particular stimuli is widely known and uncontroversial. The results do not support the authors' main conclusion: 'Given these strong indications that the effect of monocular deprivation is a form of plasticity, our observation that the effect is present in V1 calls for a reconsideration of the established idea that human adult primary sensory cortex is resilient to plastic change.' Most of the evidence cited in the Discussion that these effects reflect substantial, long-term changes in the visual system, akin to developmental plasticity, is based on other work, not the experiments in this paper.

Please see our response to item #1 and #2 in the editorial assessment.

Nonetheless, as indicated above, the study is a very nice characterization of visual system responses to monocular contrast adaptation. The fMRI results are interesting and well characterized, and the relationship between the BOLD changes and the behavioral changes make for a nice contribution.

We thank the reviewer for the constructive criticism and the important suggestions for reframing the presentation of our results.

Finally, there is one issue with regard to the BOLD results that I found puzzling. The authors measure the ocular dominance of cortical locations ('surface vertices') but do not want the reader to confuse this with the notion of ocular dominance columns. What, exactly, is the interpretation of reasonably reliable and repeatable measures of ocular preference?

Please see our response to item #3 in the editorial assessment.

Reviewer #2:[…] I have some suggestions for additional information and discussion I would like to see included:1) The translucent eye patch used to induce the 2 hours of visual deprivation allowed light to reach the retina, but prevented pattern vision. This patch did not only remove high spatial frequency and orientation information (I assume) but likely also removed phase information from the scene. Is this correct and/or relevant to the present experiments and findings? If so, can the authors discuss what implications this may have had in terms of impact on eye specific processing and brain activity measured presently.

Please see our response to item #5 in the editorial assessment.

2) In the Materials and methods, it becomes clear that the subjects were not just idly and freely reading or using a computer during the 2 hours of deprivation, as stated in the subsection “Short-term Monocular Deprivation”, but that they underwent retinotopic fMRI "immediately after application of the patch". What proportion of the 2 hours was spent with the non-deprived eye being stimulated with the high-contrast, broadband stimuli typical of retinotopic mapping? Given the clear demonstration that only 2 hours of monocular form deprivation appears to have significant impact on neural properties in the visual system, could the stimulation received by the non-deprived eye have influenced the outcome obtained here and if so, how?

Please see our response to item #6 in the editorial assessment.

3) In the fMRI flat maps shown (Figures 1B, 5A), area MT appears to have very few significant voxels. So few in fact, that one wonders how reliable the analyses performed with respect to MT are. Please discuss.

Please see our response to item #7 in the editorial assessment.

4) Are the authors certain that the area they label as MT is truly MT, and not the MT complex, or hMT+?

It is indeed hMT+ (defined by pooling the MT and MST sub-regions in the Glasser atlas), thank you for pointing this out. This is now stated in the Materials and methods section.

5) In the Results, subsection “Figure 4: Deprivation affects spatial frequency selectivity in V1”, it is claimed that prior to deprivation, the BOLD response in V1 has broad-band selectivity with peak SF around 1 cpd. However, the graph in Figure 4A suggests that the peak is closer to 0.4 to just under 1 cpd. Please fix or discuss.

Thank you, this has been revised to “peaking at intermediate SFs, between 0.4 and 1.1 cpd, similar for the two eyes”.

6) In the context of both motivating the present study (i.e. in the Introduction) and providing an example of how the present knowledge gained could be applied in the context of a clinical population, reference should be made to keratoconic patients. Much more so than amblyopia, keratoconus develops in early adulthood, well after the end of the critical period for ocular dominance plasticity. Moreover, just like the deprivation used here, it does not deprive the eye of light, but rather introduces severe optical aberrations, largely to one eye in many cases, filtering out high spatial frequency and phase information. Several excellent studies of neural adaptation to keratoconus have been published (see Sabesan, Yoon and colleagues, among others), at least some of which should be referenced in the present work.

We thank the reviewer for this suggestion. Please see our response to item #8 in the editorial assessment.

References

Lunghi C., Morrone M.C,, Secci J., Caputo R. Binocular Rivalry Measured 2 Hours After Occlusion Therapy Predicts the Recovery Rate of the Amblyopic Eye in Anisometropic Children. *Investigative Ophthalmology & Visual Science* April 2016, Vol.57, 1537-1546. doi:10.1167/iovs.15-18419